# GeoCLIP: Clip-Inspired Alignment between Locations and Images for Effective Worldwide Geo-localization

**Vicente Vivanco Cepeda, Gaurav Kumar Nayak, Mubarak Shah**

Center for Research in Computer Vision, University of Central Florida, USA
{vicente.vivancocepeda, gauravkumar.nayak}@ucf.edu; shah@crcv.ucf.edu

## Abstract

Worldwide Geo-localization aims to pinpoint the precise location of images taken anywhere on Earth. This task has considerable challenges due to the immense variation in geographic landscapes. The image-to-image retrieval-based approaches fail to solve this problem on a global scale as it is not feasible to construct a large gallery of images covering the entire world. Instead, existing approaches divide the globe into discrete geographic cells, transforming the problem into a classification task. However, their performance is limited by the predefined classes and often results in inaccurate localizations when an image's location significantly deviates from its class center. To overcome these limitations, we propose GeoCLIP, a novel CLIP-inspired Image-to-GPS retrieval approach that enforces alignment between the image and its corresponding GPS locations. GeoCLIP's location encoder models the Earth as a continuous function by employing positional encoding through random Fourier features and constructing a hierarchical representation that captures information at varying resolutions to yield a semantically rich high-dimensional feature suitable to use even beyond geo-localization. To the best of our knowledge, this is the first work employing GPS encoding for geo-localization. We demonstrate the efficacy of our method via extensive experiments and ablations on benchmark datasets. We achieve competitive performance with just 20% of training data, highlighting its effectiveness even in limited-data settings. Furthermore, we qualitatively demonstrate geo-localization using a text query by leveraging the CLIP backbone of our image encoder. The project webpage is available at: https://vicentevivan.github.io/GeoCLIP

## 1 Introduction

Image geo-localization refers to determining the geographical location where a photograph is taken from. This problem is of great significance in a wide variety of applications, such as navigation, tourism, and security. However, this task poses several challenges, especially for locations that lack distinctive landmarks or are outside tourist hotspots. Typically, image-based geo-localization mainly focuses on identifying the GPS locations of images within a specified region of interest (e.g., within a city). However, worldwide geo-localization [12, 14, 5] is not limited to any specific area but to the entire globe, thus making it significantly more challenging, and hasn't been explored extensively.

Geo-localization at the global level requires finding the locations (i.e., latitude and longitude) of images from different parts of the world. Many retrieval-based methods [28, 11, 34, 29, 32, 24, 20, 33] which perform image-to-image retrieval limit their search space to specific cities. Extending such approaches for worldwide geo-localization demands maintaining a gallery of all possible images of the entire globe to perform a match with the query image. However, creating such a large image gallery is infeasible; thus, these methods are inappropriate for retrieval at the global scale. Existing works instead use classification-based approaches [14, 26, 12, 5, 27, 8, 18] using scenes, hierarchy, queries, and segmentation. These methods divide the Earth into geographic classes (covering a particular area) and apply a linear classifier for prediction. If an image belonging to a class is far from

37th Conference on Neural Information Processing Systems (NeurIPS 2023).

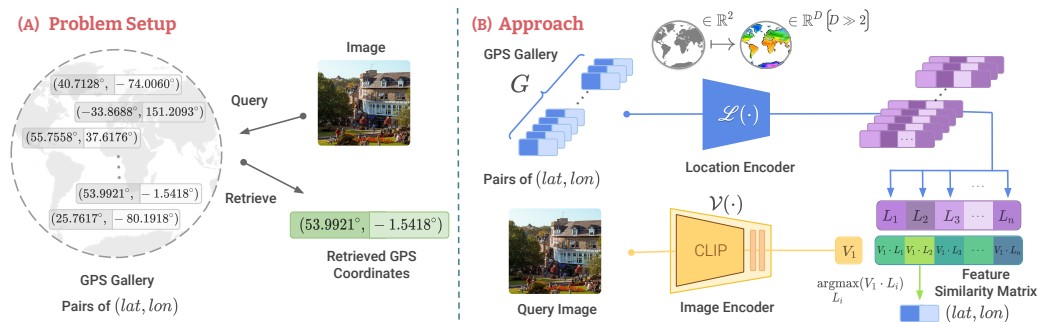

Figure 1: The objective of image-based geo-localization is to find the GPS coordinate of a given image. (A) We formulate this problem as an image-to-GPS retrieval approach where the image is a query, and the gallery is a collection of GPS coordinates. (B) We propose a novel method, GeoCLIP, having two major components: a location encoder $\mathcal{L}(\cdot)$ to obtain high-dimensional features from the 2D GPS coordinates and a CLIP-based image encoder $\mathcal{V}(\cdot)$ to obtain the image features. We perform the matching of features of a query image against a gallery of GPS embeddings, and the most similar GPS embedding is the predicted GPS coordinates.

its center, it would result in a localization error even though the classification prediction is correct. Also, the predefined classes limit the prediction to $\approx$ 21k possible locations, which is not dense enough to predict any location in the world. Moreover, these methods often need large amounts of training data which takes days to train even with high computational resources. Thus, there is a need for a new effective and efficient approach that can better model the relationships between images and their corresponding geographic locations. We summarize the problem setup and compare the major differences between existing works and our proposed method in Figure 1.

To address the above limitations of existing approaches, we propose a CLIP[15]-inspired image-to-GPS retrieval approach (named 'GeoCLIP') where we retrieve the GPS coordinates of an unseen query image by matching it with the gallery of GPS coordinates. To facilitate matching, we design location and image encoders to project GPS and image in a shared embedding space. Our location encoder transforms a given 2D GPS coordinate into a high-dimensional feature space. The traditional latitude and longitude systems have inherent distortions in representing certain areas (e.g., countries near to poles and equator have inconsistent representations). Hence, we first convert the GPS locations using equal earth projection [17] to minimize such distortion. Another challenge is to obtain a high-dimensional representation of a 2D GPS coordinate. The direct use of standard MLP leads to spectral bias, as they approximate low-frequency functions and fail to capture high-frequency details with low-dimensional inputs. Previous studies [21] have demonstrated that such spectral bias can significantly degrade the performance in various applications, particularly when capturing these details is crucial for the task. To alleviate this, we employ a positional encoding technique using Random Fourier Features (RFF) [21] before processing it through a feedforward network.

To enrich the location features, we further enforce hierarchical representation in our location encoder by capturing features at different granularities (ranging from coarse to fine-grained resolutions) and combining them to obtain a rich high-dimensional feature. Specifically, we vary the frequencies in RFF by changing sigma values using our exponential assignment strategy, enabling us to process the 2D GPS coordinates at different resolutions. We employ contrastive loss between features of the image and GPS. It is well established that the incorporation of additional negative samples enhances the performance of contrastive learning. Our gallery is a collection of GPS coordinates instead of images. Hence it is easy for us to fetch negatives by uniformly sampling the GPS coordinates from the globe. To this end, we implement a queue of GPS values to serve as artificial negatives. Our location encoder learns generic features as we find them suitable for use even beyond geo-localization. Moreover, using the CLIP [15] backbone in our image encoder helps extract meaningful features and allows us to compare GPS coordinates with text descriptions, which is not feasible in existing works. Besides this, our image-to-GPS retrieval method facilitates a much finer evaluation than previous techniques, resulting in better localization performance.

Below we summarize our major contributions as follows:

- To the best of our knowledge, we are the first to solve worldwide geo-localization task via image-to-GPS retrieval approach (GeoCLIP) by explicitly aligning the features of images to corresponding GPS locations.

- Our location encoder incorporates positional encoding with random Fourier features to efficiently encode GPS coordinates and mitigate spectral bias in MLPs. In addition, we use an exponential sigma assignment strategy to facilitate learning hierarchical features at different resolutions (Sec. 3.1.1).
- We validate the efficacy of our location encoder in an additional application, highlighting its versatility beyond geo-localization tasks (Sec. 4.5).
- Unlike existing works that only use an image as query, we also qualitatively demonstrate global geo-localization with text query using GeoCLIP (Sec. 4.4).
- We also demonstrate the effectiveness of our approach in limited data settings (Sec. 4.2).

## 2 Related Works

This section briefly discusses the relevant works related to our approach.

**Global Image Prediction:** Many previous works [27, 18, 26, 12, 14], have tackled the problem of global image geo-localization. While the most common localization papers use an image-to-image retrieval technique [16, 19, 20, 25, 34, 32], this is not yet feasible on the global scale because of the data collection that would be necessary for a global reference dataset to compare against. Instead, Weyand *et al.* [27] proposed to split the earth into individual classes to predict the location of an image. Through constructing these classes, the prediction can be relatively coarse or fine-grained based on the geographic size of the classes. Vo *et al.* [26] utilized multiple levels of geographic classes to learn different granularities, and Seo *et al.* [18] was the first to combine these levels together for an enhanced prediction. Müller *et al.* [12] additionally uses the scene labels of images to learn by having separate encoders designated to learn each scene's features. Recently Clark *et al.* [5] made use of even more hierarchies and scenes than the previous works, they also use a specialized transformer decoder to learn different features from every input image that represents each hierarchy and scene. The use of semantic segmentation was introduced by [14] in an attempt to provide additional context and understanding for the model to better geo-localize. They use two transformer encoders, one for the RGB image and the other for the segmented representation of that image, and share information between the encoders to get a better location prediction. Unlike existing global image geo-localization works, we instead take an image-to-GPS retrieval approach by directly matching the query image features to the features of GPS data.

**Learning from GPS Data:** The GPS coordinates of an image are used in [22, 10, 30, 31, 3] to provide additional context when performing classification. They construct GPS embeddings by leveraging image tags, textual descriptors, and information from geospatial databases, which are paired with image embeddings to improve the classification accuracy. We differ from this work by utilizing Random Fourier Features (RFF) [21] with MLPs. RFF is often used in NeRFs to represent the location or viewing angle for reconstructing the image or scene at that location or from that view. However, our location encoder uses RFF to represent a specific location on the earth and aligns them with the corresponding image features. We later also show that GPS embeddings of our location encoder is generic and can even be utilized for classification tasks as well, leading to further performance gains and showing its utility beyond geo-localization tasks.

**Contrastive Learning:** It is used in a wide variety of tasks in multiple deep learning-based disciplines. The basic idea is to match the features from a positive pair while pushing away from negative example features. Our method uses two different contrastive learning strategies, SimCLR[2] and CLIP[15]. SimCLR uses a base image and creates two separately augmented versions of it. It then attempts to match these two versions of the image together and push away the other images that are not from the same base. They show that this creates a robust feature representation of images. On the other hand, CLIP utilizes a large image-text database to learn a shared feature space between its image and text encoders. To establish a meaningful and distinct representation for both the text and image, they enforce similarity between the image features and the corresponding text caption. CLIP has been widely used in many tasks since its inception. We utilize the pretrained model from CLIP as a backbone to our image encoder, while our augmentation strategy is similar to SimCLR. However, instead of matching the image to another image, as in SimCLR, or to text, as in CLIP, we match it to a gallery of GPS features. The additional benefit of using CLIP backbone is the ability to compare GPS coordinates through our location encoder and text from CLIP's text encoder, which provides a very intriguing qualitative analysis that we show later (Sec. 4.4). Now, we explain our proposed approach in detail in the next section.

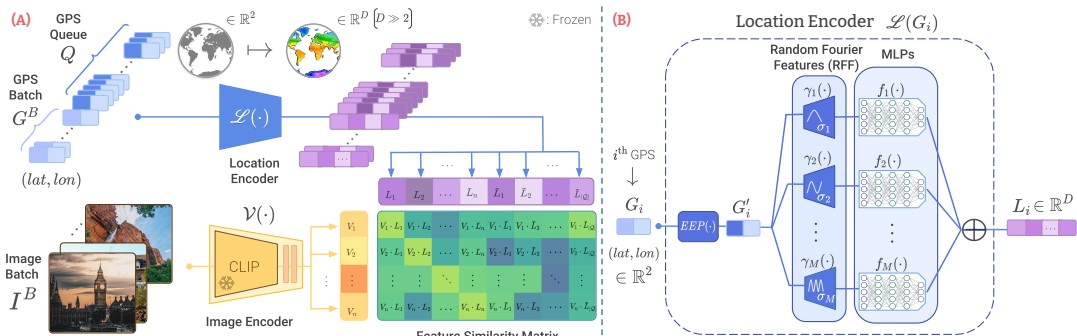

Figure 2: (A) Given a batch of GPS coordinates ($G^B$), we generate additional negatives $Q$ using our dynamic queue strategy. Each image in batch ($I^B$) is processed using our image encoder $\mathcal{V}(\cdot)$ with CLIP[15] backbone. Similarly, we process the GPS coordinates using our location encoder $\mathcal{L}(\cdot)$. We train our model to align image features with corresponding GPS embeddings using contrastive loss over feature similarity matrix. (B) $\mathcal{L}(\cdot)$ transforms 2D coordinates $G_i$ into $G'_i$ using equal earth projection (EEP)[17]. Then we obtain hierarchical representations of $G'_i$ using RFF [21] and MLPs, aggregated to obtain a rich high-dimensional representation $L_i$.

## 3  Proposed Approach

**Worldwide Image Geo-localization Setup**: Given the access to training dataset $D_{train} = \{(I_n, G_n)\}_{n=1}^{N}$ containing pairs of image $I_n$, and GPS coordinate $G_n$. Our goal is to train a world-wide model $W$ using samples from $D_{train}$ and then use the trained model $W$ to predict the location of unseen $K$ query images of test dataset $D_{test}$. Specifically, our objective is to accurately predict the latitude and longitude of query images, $G_k^Q = W(I_k^Q), \forall k \in [1..K]$ where $I_k^Q \in D_{test}$. The images in the query set $I^Q$ can belong to any geographical location of Earth, making it a challenging setup.

We formulate this problem as a retrieval problem. Traditionally, retrieval involves matching a query image with images from the gallery, and the GPS coordinates of the closest gallery image would be the desired output. This approach is often used when the search space is limited to specific cities. However, it has limitations when applied to retrieval at the global scale, which requires the construction of a gallery containing images covering the entire world. Instead, it is much easier to build a gallery of GPS coordinates. Hence, geo-localization can be framed as an image-to-GPS matching problem where the closest GPS match would be the location of the query image.

In order to perform matching, the features of the image and GPS location need to be extracted. Hence, we have two major components in our architecture, i.e., Location Encoder $\mathcal{L}(\cdot)$ and Vision Transformer $\mathcal{V}(\cdot)$ as Image Encoder. Now, we explain the model architecture containing details of both the encoders, followed by the training and evaluation strategies.

### 3.1  Model architecture

The architecture of our method GeoCLIP is shown in Figure 2.

**Image Encoder**: The CLIP [15] model has demonstrated strong generalization performance across a wide range of computer vision tasks. Motivated by it, our image encoder contains a Vision Transformer, which is a pre-trained CLIP ViT[6]-L/14 model. We use it as a backbone and keep it frozen. To adapt this backbone for our geo-localization task, we add two linear trainable layers with dimensions of $h_1$ and $h_2$, respectively. These additional layers are employed to fine-tune the learned representation for our task. Hence, our overall image encoder $\mathcal{V}(\cdot)$ extracts the features $V_i = \mathcal{V}(I_i), \forall i \in [1 \ldots N]$ and $I_i \in D_{train}$.

Next, we explicitly encode the GPS coordinates, which has not been tried for the geo-localization task. We now explain our location encoder in detail, which encodes the GPS coordinates of a location.

### 3.1.1  Location Encoder

Encoding 2D GPS coordinates (i.e., $G_i \in \mathbb{R}^2, \forall i \in [1 \ldots N]$) to a high-dimensional representation ($\mathbb{R}^D$) where $D \gg 2$, is a challenging task. A simple use of standard MLPs for such encoding suffers from spectral bias [21] (unable to capture high-frequency details). Hence, to handle this challenge and effectively perform this encoding, we employ different components in our encoder, i.e., representing

GPS coordinates using equal earth projection[17], using positional encoding through random Fourier features, and enforcing hierarchical representation at various scales. We explain each of them below:

**Equal Earth Projection (EEP)**: To accurately represent 2D GPS coordinates and minimize distortions inherent in standard coordinate systems (i.e., countries closer to the poles are overrepresented in traditional latitude and longitude systems), we employ the Equal Earth Projection (EEP)[17]. By utilizing EEP, we counterbalance the distortions and ensure a more accurate representation of the Earth's surface. Given $G_i \in \mathbb{R}^2$ (in radians), we transform it into $G_i'$ by applying the EEP as below:

$$G_i'^{lat} = \frac{2\sqrt{3}G_i^{lon}cos\theta}{3(9\,P_4\,\theta^8 + 7\,P_3\,\theta^6 + 3\,P_2\,\theta^2 + P_1)}; \; G_i'^{lon} = P_4\,\theta^9 + P_3\,\theta^7 + P_2\,\theta^3 + P_1\,\theta \quad (1)$$

where $\sin\theta = \frac{\sqrt{3}}{2}\sin G_i^{lat}$, $P_1 = 1.340264$, $P_2 = -0.081106$, $P_3 = 0.000893$, $P_4 = 0.003796$. After applying the EEP, we scale the resulting longitude in the range $-1$ to $1$, and the latitude values are scaled proportionally.

**Random Fourier Features (RFF)**: To capture rich high-frequency details from low-dimensional GPS input, we apply a sinusoidal-based positional encoding technique to $G_i'$. Specifically, we leverage random Fourier features (RFF)[21] for this positional encoding. We construct an encoding layer that determines the range of frequencies to be encoded. We limit the frequencies using a fixed matrix $\mathbf{R}$, whose entries are sampled from a Gaussian distribution with the standard deviation ($\sigma$). The matrix $\mathbf{R}$ is set at the beginning of training and remains unchanged throughout the training process. The RFF operation $\gamma(\cdot)$ encodes GPS coordinate $G_i'$ as $\gamma(G_i') = [\cos(2\pi\mathbf{R}G_i'), \sin(2\pi\mathbf{R}G_i')]^\mathrm{T}$, where the entries of a $m^{th}$ row and $n^{th}$ column of matrix $\mathbf{R}$ are $\mathbf{r}_{m,n} \sim \mathcal{N}(0, \sigma)$.

**Hierarchical Representation**: To construct a hierarchical representation capable of capturing information at varying scales, we vary the frequency in RFF to obtain multiple high-dimensional representations spanning from coarse to fine-grained. As the frequency range in RFF depends on the $\sigma$ value, we propose an exponential assignment strategy for choosing the $\sigma$ values in our location encoder. Specifically, for a given a range of $\sigma$ values (i.e., $[\sigma_{min}, \sigma_{max}]$), we obtain the $\sigma$ values for $M$ level hierarchy using the following:

$$\sigma_i = 2^{\log_2(\sigma_{min})+(i-1)(\log_2(\sigma_{max})-\log_2(\sigma_{min}))/(M-1)}, \quad \forall i \in \{1 \ldots M\}. \quad (2)$$

This exponential assignment of $\sigma$ values enables our location encoders $\mathscr{L}(\cdot)$ to effectively process the input 2D coordinates at different resolutions. Consequently, this hierarchical setting allows $\mathscr{L}(\cdot)$ to specialize in capturing features of a particular location at various scales. It enforces our model to efficiently learn a rich understanding of spatial information, making it suitable for a wide range of geospatial deep learning tasks.

Next, we pass the encoded hierarchical features through a feedforward MLP. Each MLP $f_i$ independently processes the outputs of RFF and yields a vector $\mathbf{v}_i \in \mathbb{R}^d$. We then perform element-wise addition of these resulting vectors to obtain a joint representation $\mathbf{v} = \sum_{i=1}^{M} \mathbf{v}_i$.

Below we summarize the overall operations performed by our location encoder $\mathscr{L}(\cdot)$ for obtaining rich high-dimensional encoded features ($L_i$) of a GPS 2D coordinate $G_i$:

$$L_i = \mathscr{L}(G_i) = \sum_{i=1}^{M} f_i(\gamma(EEP(G_i), \sigma_i)) \quad (3)$$

### 3.2 Model Training

We train our model using a contrastive learning scheme (similar to [15]) where we maximize the similarity of features of an image $I_i$ (i.e., $V_i$) with its corresponding location features of GPS coordinate $G_i$ (i.e., $L_i$) and minimize its similarity with other GPS coordinates in a batch. To further improve our learned representation, we also adopt a training scheme similar to SimCLR [2].

We apply a set of random augmentations ($\mathcal{A}$) to each image in a batch and create $P$ views of an image to improve the quality of learned representations and promote better generalization. We employ the same set of augmentations used in the SimCLR. For each augmented image $I_{ij}$ ($i^{th}$ sample of $D_{train}$ and $j^{th}$ view, we also create variation in its corresponding GPS coordinates $G_i$ by injecting random noise $\eta_j$ to it (denoting it by $G_ij$) which helps to enforce spatial smoothness in the location encoder. The noise $\eta_j$ is sampled from a Gaussian distribution with a standard deviation of $\sigma_\eta$ (i.e.,

$\eta \sim \mathcal{N}(0, \sigma_\eta^2)$. This added noise encourages the location encoder to learn more robust representations that are less sensitive to small variations in the GPS coordinates.

At any point during training, our location encoder $\mathcal{L}(\cdot)$ can obtain the embedding of any GPS location $g$ (i.e., $L_g$) without requiring a corresponding image. We exploit this capability to generate additional negative samples during training to improve the learned representation with contrastive learning. We append a queue $Q$ of GPS locations with a fixed length $S$ to every batch of training data $D_{train}$. During training, we generate embeddings for these GPS coordinates ($\tilde{L}$). We also inject noise ($\eta' \sim \mathcal{N}(0, \sigma_{\eta'}^2)$) to $\tilde{L}$, such that $\sigma_{\eta'} \geq \sigma_\eta$. This additional queue of GPS embeddings notably improves the performance of our model, particularly at smaller scales (refer Table 3(a)).

Hence, for an $i^{th}$ sample of a batch ($I_i^B, G_i^B \in D_{train}$) having $P$ augmented views with temperature $\tau$, our overall training objective is to minimize the following loss:

$$\mathcal{L}_i = -\sum_{j=1}^{P} \log \frac{\exp\left(V_{ij} \cdot L_{ij}/\tau\right)}{\sum_{i=0}^{|B|} \exp(V_{ij} \cdot L_{ij}/\tau) + \sum_{i=0}^{S} \exp(V_{ij} \cdot \tilde{L}_i/\tau)}. \tag{4}$$

After the model parameters are updated on a batch of $D_{train}$ with batch size $|B|$, we update the queue $Q$ by replacing its oldest coordinates with GPS coordinates from the current batch $\hat{G}^B$ where $|B| \ll S = |Q|$, ensuring an up-to-date reflection of the most recent GPS coordinates encountered (naming it as 'dynamic queue' strategy).

**Evaluation**: By modeling the Earth as a continuous function, our location encoder can generate an embedding for any GPS coordinate on the planet. Hence, our method allows us to directly compare the embedding of a query image with the embeddings of any chosen GPS location, effectively eliminating the constraint of working within a predetermined set of classes.

Unlike existing works that create classes using the knowledge of the training set, we create a gallery of possible GPS coordinates during the evaluation phase by randomly sampling from the training set. The other ways for constructing the GPS gallery are discussed in the supplementary. To obtain the GPS predictions ($G_k^Q$) for a $k^{th}$ image in the unseen query set ($I_k^Q$), we compare the query image's embedding with the embeddings of GPS coordinates in our gallery. The GPS coordinate with the highest similarity to the query image's embedding indicates the most probable location for the image (i.e., $G_k^Q$ is $\arg\max_{G_i^Q \in D_{test}} \mathcal{V}(I_k^Q) \cdot \mathcal{L}(G_i^Q)$).

## 4 Experiments

**Datasets and Evaluation details**: We perform experiments on publicly available benchmark datasets to have a fair comparison with existing works. For training, we use the MediaEval Placing Tasks 2016 (MP-16) dataset [9], which consists of 4.72 million geotagged images from Flickr [1]. We test our trained model on several different datasets: Im2GPS3k [7], the recently introduced Google World Streets 15K dataset (GWS15k) [5], and YFCC26k [23]. During testing, we conduct image-to-GPS retrieval where the query image from the test datasets is matched against a GPS gallery of 100K (Im2GPS3k) and 500K (GWS15k) coordinates. The performance of GeoCLIP on different gallery sizes is put in supplementary. In our evaluation, similar to [5], we also uses a Ten Crop (5 different crops of an image and their flipped counterparts) method where the predictions on these crops are averaged to make a joint prediction. We report our results using a threshold metric, similar to the protocol followed in previous works. Specifically, for each query image, we compute the Geodesic distance between GPS coordinates predicted by our model and the respective ground truth. We calculate how many of them (in %) fall within the distance thresholds (1km, 25km, 200km, 750km, and 2500km) and report average performance of model over three runs.

**Implementation details**: Each MLP $f_i$ used in our location encoder $\mathcal{L}(\cdot)$ is composed of an input layer with 512 dimensions, four hidden layers (each with ReLU activation and 1024 dimensions), and an output layer of 512 dimensions. The RFF of the positional encoding layers have an input size of 2 dimensions and an output size of 512 dimensions. The dynamic queue ($|Q|$ as 4096) is initialized randomly with values drawn from a uniform distribution covering the range of latitude and longitude. We add gaussian noise to GPS coordinates with $\sigma_\eta$ and $\sigma_{\eta'}$ as 150 and 1000 respectively. Further ablations on the noise and queue length are put in supplementary. We use three hierarchies ($M$ as 3) with $\sigma_{min}$ as $2^0$, and $\sigma_{max}$ as $2^8$ respectively (refer Sec. 4.3). Our image encoder $\mathcal{V}(\cdot)$ utilizes OpenAI's pre-trained CLIP ViT-L/14 as backbone with two trainable linear layers $h_1$ and $h_2$ having dimensions of 768 and 512 respectively. We perform image augmentations similar to SimCLR.

**Training details**: Our GeoCLIP model is trained using Adam optimizer with a learning rate of $3 \times 10^{-5}$ and a weight decay of $1 \times 10^{-6}$. We use a learning rate of $3 \times 10^{-4}$ to train classifiers with the CLIP backbone. We employ a step decay learning rate scheduler with a gamma of $0.87$ and a step size of one epoch. The model is trained until convergence, which typically occurs around tenth epoch. We use batch size $|B|$ as $512$ when training on full dataset. For data-efficient settings with $20\%$, $10\%$, and $5\%$ of the data, we use $|B|$ as $256$, $256$, and $128$ respectively. The learnable temperature parameter $\tau$ is initialized to $0.07$ (refer eq. 4). The CLIP backbone is frozen during training while location encoder and $h_1$ and $h_2$ layers are fully trainable. We train our models on an NVIDIA Ampere GPU along with 12 CPUs. Our code is in PyTorch [13] and is shared in supplementary.

## 4.1 Comparison with State-of-the-art methods

Table 1: We compare the performance of GeoCLIP with the state-of-the-art methods on (a) Im2GPS3k [7] and (b) GWS15k [5] datasets. Our method yields consistent gains across datasets and different distance thresholds.

(a) Results on the Im2GPS3k [7] dataset

| Method | Street 1 km | City 25 km | Region 200 km | Country 750 km | Continent 2500 km |
|---|---|---|---|---|---|
| [L]kNN, $\sigma = 4$ [26] | 7.2 | 19.4 | 26.9 | 38.9 | 55.9 |
| PlaNet [27] | 8.5 | 24.8 | 34.3 | 48.4 | 64.6 |
| CPlaNet [18] | 10.2 | 26.5 | 34.6 | 48.6 | 64.6 |
| ISNs [12] | 10.5 | 28.0 | 36.6 | 49.7 | 66.0 |
| Translocator [14] | 11.8 | 31.1 | 46.7 | 58.9 | 80.1 |
| GeoDecoder [5] | 12.8 | 33.5 | 45.9 | 61.0 | 76.1 |
| Ours | **14.11** | **34.47** | **50.65** | **69.67** | **83.82** |

(b) Results on the recent GWS15k [5] dataset

| Method | Street 1 km | City 25 km | Region 200 km | Country 750 km | Continent 2500 km |
|---|---|---|---|---|---|
| ISNs [12] | 0.05 | 0.6 | 4.2 | 15.5 | 38.5 |
| Translocator [14] | 0.5 | 1.1 | 8.0 | 25.5 | 48.3 |
| GeoDecoder [5] | **0.7** | 1.5 | 8.7 | 26.9 | 50.5 |
| Ours | 0.6 | **3.1** | **16.9** | **45.7** | **74.1** |

We perform a comparative analysis of GeoCLIP against worldwide Geo-Localization benchmarks, Im2GPS3k and Google World Streets 15k (GWS15k). The results are shown in Table 1. In all the metrics, our method surpasses the previous state-of-the-art (SOTA) performance on Im2GPS3k, achieving improvements of $+1.31\%$, $+0.97\%$, $+3.95\%$, and $+8.67\%$ in the 1km, 25km, 200km, 750km, and 2500km thresholds respectively. The results on another dataset (YFCC26k) are put in supplementary, where we also observe a similar trend.

Notably, our approach exhibits a large gain on the more challenging GWS15k dataset, surpassing the previous SOTA model with significant accuracy improvements of $+1.6\%$, $+8.2\%$, $+18.8\%$, and $+23.6\%$ in the 25km, 200km, 750km, and 2500km thresholds respectively. Our performance is nearly double the accuracy of the previous state-of-the-art model. The GWS15k contains samples that are uniformly sampled across the Earth and are not biased towards any specific geographic location. Moreover, the images in this dataset have a large distribution shift compared to the training set, making the geo-localization task tough and challenging. Our substantial improvement can be attributed to the better embeddings achieved by our encoders. Specifically, the hierarchical nature of the location encoder helps focus on multiple resolutions of a location simultaneously, and its alignment with corresponding out-of-distribution images gets further enhanced with the robust features obtained by the CLIP backbone of our image encoder.

## 4.2 Performance of GeoCLIP in Limited Data settings

The worldwide geo-localization methods assume access to a training set containing millions of data samples. Training on such large scale datasets is often highly time-consuming, even with substantial computational resources at hand, taking days to complete. Moreover, practical limitations may arise, such as restricted access to data due to privacy concerns or proprietary rights held by companies. Hence, we also investigate the potential of our GeoCLIP method in limited-data settings. In the Figure 3, we present a comprehensive analysis of our method's performance on Im2GPS3k dataset across various data settings: $20\%$, $10\%$, $5\%$ of the full dataset. Results on other datasets are provided in the supplementary. We can observe that our method maintains a high level of performance, closely approximating the results achieved with the full dataset, even as the available data decreases exponentially. Notably, even with a mere $20\%$ of the data, the performance of GeoCLIP drops by only $0.0\%$, $0.5\%$, $1.6\%$, $1.9\%$, and $1.0\%$ on 2500km , 750km, 200km, 25km, and 1km distance thresholds respectively. These results highlight the efficiency of GeoCLIP in limited-data scenarios, emphasizing its potential to address practical challenges in worldwide geo-localization.

To further demonstrate the better efficacy of our method in limited data settings, we perform the same experiment with classification approach [12] (ISNs). We summarize the comparison of our method

with the classification approach on different quantities of limited training data in Table 2. It can be easily observed that the classification method's performance degrades much faster than our approach when the amount of training data is reduced, highlighting our method to be more data-efficient.

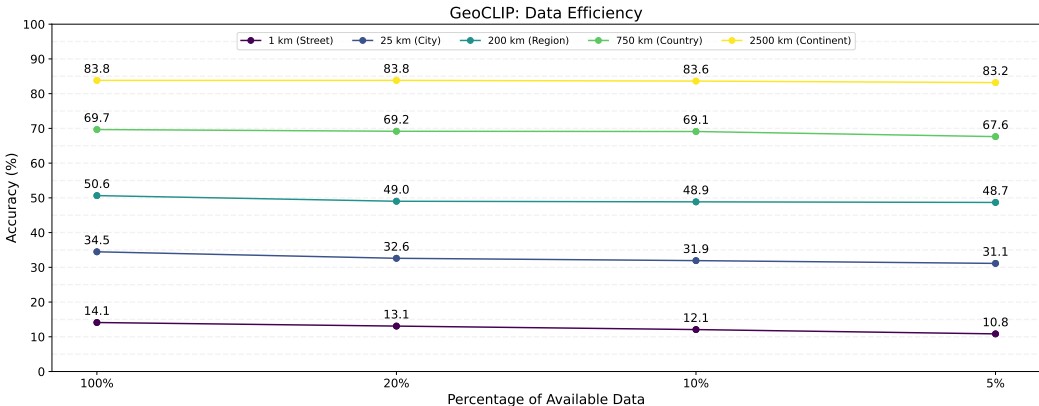

Figure 3: We vary the amount of training data from $100\%$ to as low as $5\%$. GeoCLIP achieves respectable accuracy even with only $5\%$ of the training data, demonstrating its utility in limited data scenarios.

Table 2: Comparison of data efficiency of GeoCLIP with respect to a classification approach when the amount of training data is reduced drastically. The value inside brackets on each row shows the performance degradation compared to the respective method's performance with $100\%$ training data.

| Method | Im2GPS3k (Amount of Training Data) | | | |
|---|---|---|---|---|
| | 100% | 20% | 10% | 5% |
| 2500 km (GeoCLIP, Ours) | 83.8 | 83.8 ($\downarrow$0.0) | 83.6 ($\downarrow$0.2) | 83.2 ($\downarrow$0.6) |
| 2500km (ISNs [12]) | 66.0 | 50.1 ($\downarrow$15.9) | 48.7 ($\downarrow$17.3) | 43.9 ($\downarrow$22.1) |
| 750km (GeoCLIP, Ours) | 69.7 | 69.2 ($\downarrow$0.5) | 69.1 ($\downarrow$0.6) | 67.6 ($\downarrow$2.1) |
| 750km (ISNs [12]) | 49.7 | 31.9 ($\downarrow$17.8) | 29.2 ($\downarrow$20.5)) | 25.1 ($\downarrow$24.6) |
| 200km (GeoCLIP, Ours) | 50.6 | 49.0 ($\downarrow$1.6) | 48.9 ($\downarrow$1.7) | 48.7 ($\downarrow$1.9) |
| 200km (ISNs [12]) | 36.6 | 20.0 ($\downarrow$16.6) | 17.4 ($\downarrow$19.2) | 14.3 ($\downarrow$22.3) |
| 25km (GeoCLIP, Ours) | 34.5 | 32.6 ($\downarrow$1.9) | 31.9 ($\downarrow$2.6) | 31.1 ($\downarrow$3.4) |
| 25km (ISNs [12]) | 28.0 | 14.1 ($\downarrow$13.9) | 11.9 ($\downarrow$16.1) | 9.6 ($\downarrow$18.4) |
| 1km (GeoCLIP, Ours) | 14.1 | 13.1 ($\downarrow$1.0) | 12.1 ($\downarrow$2.0) | 10.8 ($\downarrow$3.3) |
| 1km (ISNs [12]) | 10.5 | 5.1 ($\downarrow$5.4) | 4.5 ($\downarrow$6.0) | 3.2 ($\downarrow$7.3) |

## 4.3 Ablations

**Effective encoding of raw GPS coordinates**: The direct use of an individual MLP to encode GPS coordinates into feature vectors serves as a baseline. As shown in Table 3(a), the MLP performs well at larger scales, but it stumbles at finer scales due to distortions in the standard coordinate system and its own spectral bias. To handle the distortions, we apply the Equal Earth Projection (EEP), which boosts performance at 1km, 25km, and 200km thresholds by $+0.1\%$, $+6.07\%$, and $+3.93\%$, respectively. However, the encoding of fine-grained details remains a challenge. To address this, we introduce an additional encoding layer using Random Fourier Features ($\sigma$ as $2^4$). This layer enables the MLP to learn high-frequency functions necessary to distinguish features at finer scales, like streets and cities. This modification nearly doubles the accuracy at 1km and 25km, with increases of $+5.14\%$ and $+14.95\%$, respectively. When we include harder negatives via our dynamic queue, improving performance across all scales, with a notable increase of $+4.47\%$ at the 1km threshold.

**Importance of Hierarchical Learning**: We experiment with different $\sigma$'s in eq. 2 ($2^0$, $2^4$, $2^8$) to investigate their influence across various scales. With a single accuracy metric, one might typically seek an optimal sigma. However, we're optimizing for five scale metrics (1km, 25km, 200km, 750km, and 2500km), making the task more difficult. Our findings, highlighted in Table 3(b) and further elaborated in the supplementary material, reveal an interesting trade-off: smaller $\sigma$ excels at larger scales but has issues with smaller scales, while larger $\sigma$ increases fine-grained accuracy but decreases on larger scales. Building upon this observation, we combine the representations from the encoders with varying $\sigma$s. By merging the outputs from the different encoders, we create a joint hierarchical representation that maximizes the location encoder's capabilities across all scales. This method outperforms the use of individual $\sigma$ at all scales, effectively addressing the varying performance issue related to different $\sigma$s, thereby endorsing the need for hierarchical representation in our architecture.

Table 3: We perform ablations on Im2GPS3k to show the importance of different components of our location encoder. (a) Our encoding of GPS coordinates using Equal Earth Projection (EEP), Random Fourier Features(RFF), and Dynamic Queue (DQ) strategy led to significant gains compared to the MLP alone, especially in challenging localization within 1km and 25 km. (b) Our GeoCLIP approach with hierarchies (GeoCLIP w/ H) leads to further gains and even performs much better than the respective classification with hierarchies C (w/ H) approach. The improvements with respect to baseline in (a) and C (w/ H) in (b) are shown in brackets and highlighted in blue.

(a) Encoding GPS coordinates

| Method | Street 1 km | City 25 km | Region 200 km | Country 750 km | Continent 2500 km |
|---|---|---|---|---|---|
| MLP (Baseline) | 0.13 | 11.01 | 41.68 | 67.83 | 83.28 |
| +EEP | 0.23 | 17.08 | 45.61 | 68.8 | 82.98 |
| +EEP +RFF | 5.37 | 32.03 | 49.42 | 68.4 | 82.98 |
| +EEP +RFF +DQ | 9.84 | 33.1 | 49.28 | 69.07 | 83.35 |
| | (+9.71) | (+22.09) | (+7.6) | (+1.24) | (+0.07) |

(b) Benefits of hierarchical learning

| Method | Street 1 km | City 25 km | Region 200 km | Country 750 km | Continent 2500 km |
|---|---|---|---|---|---|
| C (w/ H) | 11.4 | 34.3 | 49.4 | 67.4 | 80.8 |
| GeoCLIP (Coarse) | 1.07 | 24.66 | 49.18 | 69.47 | 83.75 |
| GeoCLIP (Medium) | 9.84 | 33.1 | 49.28 | 69.07 | 83.35 |
| GeoCLIP (Fine) | 13.61 | 33.43 | 47.78 | 65.1 | 80.95 |
| GeoCLIP (w/ H) | **14.11** | **34.47** | **50.65** | **69.67** | **83.82** |
| | (+2.71) | (+0.17) | (+1.25) | (+2.27) | (+2.02) |

## 4.4 Qualitative Results: Geo-localization using Text-based query

We demonstrate GeoCLIP's ability to process text queries, highlighting the semantic richness of our model. As our image encoder has a pre-trained CLIP backbone (trained to match visual and text features), our location encoder gets inherent alignment with CLIP's text features, enabling us to map textual descriptions to geographical coordinates.

This capability allows us to assign a geographical context to any given text. Specifically, we can take any text, generate its embedding using CLIP text encoder, process it through our layers $h_1$ and $h_2$, and calculate its similarity with a set of GPS embeddings. This results in a similarity map, demonstrating the geographical distribution of the text query's concept. For instance, using "Desert" as a text query, we compute the cosine similarity between the text query's embedding and the GPS embeddings. The resulting similarity map, (Figure 4) reveals the spatial distribution of desert regions worldwide.

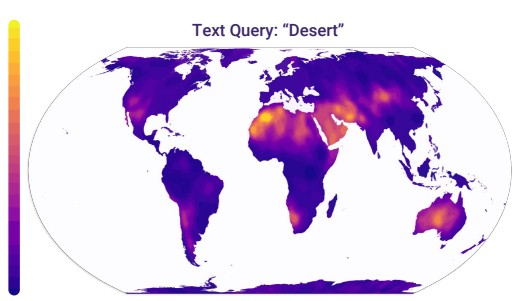

Text Query: "Desert"

| Method | Classifier | mAP |
|---|---|---|
| Visual (V) | visual | 0.234 |
| OneHot [3] | geo | 0.066 |
| HashTag [22] | geo | 0.163 |
| TagFeature [10] | geo | 0.161 |
| GPS2Vec onehot [30] | geo | 0.130 |
| GPS2Vec [30] | geo | 0.182 |
| GPS2Vec+ [31] | geo | 0.210 |
| **GeoCLIP (Ours)** | **geo** | **0.249** |
| OneHot [3] + V | fusion | 0.238 |
| HashTag [22] + V | fusion | 0.261 |
| TagFeature [10] + V | fusion | 0.260 |
| GPS2Vec Onehot [30] + V | fusion | 0.277 |
| GPS2Vec [30] + V | fusion | 0.300 |
| Liao et al. [10] | fusion | 0.347 |
| GPS2Vec+ [31] + V | fusion | 0.348 |
| **GeoCLIP (Ours) + V** | **fusion** | **0.362** |

Figure 4: Qualitative demonstration: Geo-localization with text query using GeoCLIP. Our location can differentiate between geographical coordinates and associate semantic meaning from text to geographical space, providing a geographical context to textual descriptions.

Table 4: Comparison of methods utilizing GPS data for image classification on the geotagged NUSWIDE Dataset. Our method outperforms existing methods using our GPS features with or without visual features.

## 4.5 Utility of our Location Encoder beyond Geo-localization

We investigate the utility of GeoCLIP's location encoder in improving multi-class image classification task. We perform experiments on the geo-tagged NUS-WIDE dataset [4] with the same train and test splits as used in [31]. We conduct two experiments: 1) Classifying an image using only its GPS location features, and 2) Classifying an image using a combination of its GPS location features and visual features.

We obtain GPS features from our pretrained location encoder (which is kept frozen), which are normalized and fed through a Multilayer Perceptron (MLP) with a single hidden layer of 2048 dimensions, followed by a dropout layer for regularization. For classification incorporating visual context, we adopt the Bag-of-Visual-Words (BovW) representation based on SIFT descriptors as the visual feature for a fair comparison with existing works. We normalize location and visual features, concatenate them, and then pass them through the MLP.

Our results in Table 4 indicate that GeoCLIP achieves state-of-the-art performance on both approaches. This is particularly significant, as it highlights that our location encoder has learned a semantically rich representation across the Earth, being able to generalize beyond geo-localization and can therefore be extended to assist other location-aware methods to improve performance without any significant changes to their architecture.

### 4.6 GeoCLIP with Prior Geographical Knowledge

In practical applications of geo-localization, users often possess some level of prior geographical context. For instance, while the exact GPS coordinates of an image may be unknown, the general originating area, such as the continent, country, state, or city, may be known. This additional information can effectively narrow the search space, thereby improving the model's accuracy due to the reduced number of potential locations for consideration. To assess GeoCLIP's adaptability in these situations, we perform an experiment where the gallery of potential GPS coordinates is limited within a predefined radius of known geographical knowledge of each image. We choose the radius as per the assumed level of prior knowledge (higher search radius when coarse information is known and smaller radius with the finer knowledge): for a continent, coordinates within 2500km are taken; for a country, within 750km; for a state, within 200km; and for a city, within 25km. Results across these different levels of granularity are shown in Table 5.

Table 5: Performance of GeoCLIP when the granularity of search space is varied. The performance improves as we search from coarse to finer granularity leveraging prior knowledge.

| Granularity | Street 1 km | City 25 km | Region 200 km | Country 750 km | Continent 2500 km |
|---|---|---|---|---|---|
| Continent | 15.02 | 37.70 | 55.39 | 78.11 | 100.0 |
| Country | 15.85 | 41.18 | 63.33 | 100.0 | 100.0 |
| State | 18.82 | 52.12 | 100.0 | 100.0 | 100.0 |
| City | 26.15 | 100.0 | 100.0 | 100.0 | 100.0 |

The performance of image-to-GPS retrieval improves as we search from coarse to finer granularity, which is quite intuitive. For instance, if we have prior knowledge of the city, it's much easier to geo-localize the true location than to have prior knowledge about the continent. The model's capacity to adapt to restricted search spaces affirms its applicability in real-world scenarios where partial geographic context is commonly available.

## 5 Conclusion

We addressed the challenge of worldwide geo-localization by formulating it as an image-to-GPS retrieval problem. We evaluated the efficacy of our method (GeoCLIP) not only in full training data scenarios but also in limited data settings. The performance of GeoCLIP was competitive even when the training data was reduced significantly. GeoCLIP could even perform geo-localization using text queries by leveraging the CLIP[15] backbone of our image encoder. Moreover, our location encoder (using equal earth projection[17], RFF[21], and MLPs) better encoded GPS coordinates compared to the direct use of MLPs. However, coarse $\sigma$ or a fine-grained $\sigma$ used in RFF performed better in localizing either large or small areas. Our hierarchical learning overcame this issue by capturing the location features at various scales, leading to a further performance boost. Also, our location encoder is not tied to geo-localization and helps to aid in classification problems. In the future, we plan to further investigate the utility of our location encoder in assisting other computer vision problems. **Limitations**: As we use a CLIP-based backbone in our image encoder, precomputing the image features on the large training dataset (each image of dimension $224 \times 224 \times 3$) is time-consuming. But once the features are precomputed, the training of our GeoCLIP model is relatively fast. We also observed that a single $\sigma$ value used in RFF is insufficient to perform well on both small and large areas. However, our hierarchical learning is a promising direction to overcome this issue. **Broader Impacts**: Our image-based geo-localization can act as a substitute for GPS-based localization in scenarios where GPS signals are unreliable, blocked, or inaccurate. Combining image-based geo-tagging with GPS-based positioning systems can help to improve stability and safety in applications like autonomous driving and navigation. Also, it can benefit digital forensics, but avoiding its unethical use that could mislead location detection systems or violate privacy is important. More discussion on ethical issues and possible mitigation approaches are provided in the supplementary.

## Acknowledgments and Disclosure of Funding

This work was supported by the US Army contract W911NF-2120192. We would like to extend our gratitude to all the reviewers for their valuable suggestions. We would also like to thank Brandon Clark for his helpful discussions on this project.

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
