# *Supplementary for*: "GeoCLIP: Clip-Inspired Alignment between Locations and Images for Effective Worldwide Geo-localization"

**Vicente Vivanco Cepeda,  Gaurav Kumar Nayak,  Mubarak Shah**

Center for Research in Computer Vision, University of Central Florida, USA
{vicente.vivancocepeda, gauravkumar.nayak}@ucf.edu; shah@crcv.ucf.edu

We organize our supplementary document as follows:

1. Results on additional dataset

2. Results for limited data settings on YFCC26k and GWS15k datasets

3. Additional Ablations

    (a) Gallery Size

    (b) Queue Length

    (c) $\sigma_\eta$ for Batch GPS noise

    (d) $\sigma_{\eta'}$ for Queue GPS noise

    (e) $\sigma$ for Random Fourier Features

    (f) Number of hierarchies ($M$)

4. Different selection choices for GPS Gallery Construction

    (a) Evenly Spaced GPS Coordinates

    (b) Test Set GPS Coordinates

5. Analysis of Runtime and Memory Footprint

6. Motivations for using Pretrained CLIP as Image encoder Backbone

7. Qualitative Demonstration

    (a) Hierarchical learning in our location encoder $\mathscr{L}(\cdot)$

    (b) GeoCLIP with Image Query

    (c) Distribution of correct predictions of GeoCLIP on different datasets

    (d) GeoCLIP with Text Query

8. Discussion on Ethical Issues and Possible Mitigation

37th Conference on Neural Information Processing Systems (NeurIPS 2023).

# 1 Results on additional dataset

In section 4.1 of the main paper, we demonstrated the performance of our GeoCLIP method on Im2GPS3k [2] and GWS15k [1] datasets and compared them with the state-of-the-art methods. Here, we perform experiments on another dataset YFCC26k [6]. The results are provided in Table 1.

Table 1: Results on YFCC26k [6] dataset

| Method | Street 1 km | City 25 km | Region 200 km | Country 750 km | Continent 2500 km |
|---|---|---|---|---|---|
| PlaNet [7] | 4.4 | 11.0 | 16.9 | 28.5 | 47.7 |
| ISNs [4] | 5.3 | 12.3 | 19.0 | 31.9 | 50.7 |
| Translocator [5] | 7.2 | 17.8 | 28.0 | 41.3 | 60.6 |
| GeoDecoder [1] | 10.1 | 23.9 | 34.1 | 49.6 | 69.0 |
| Ours | **11.61** | 22.19 | **36.69** | **57.47** | **76.02** |

We can observe that GeoCLIP achieves state-of-the-art performance on the YFCC26k dataset in the majority of distance threshold metrics, with $+1.51\%$, $+2.59\%$, $+7.87\%$, and $+7.02\%$ improvements in accuracy on the 1km, 200km, 750km, and 2500km respectively. This result highlights that GeoCLIP performs well across datasets, being useful across different data distributions.

# 2 Results for limited data settings on YFCC26k and GWS15k datasets

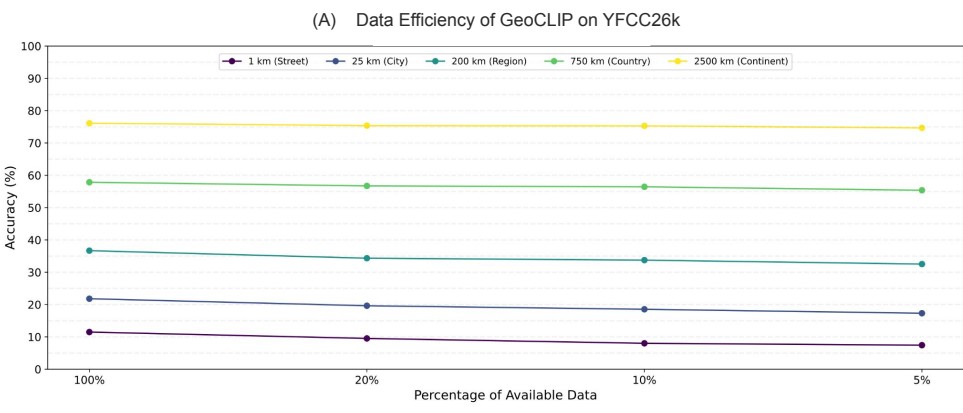

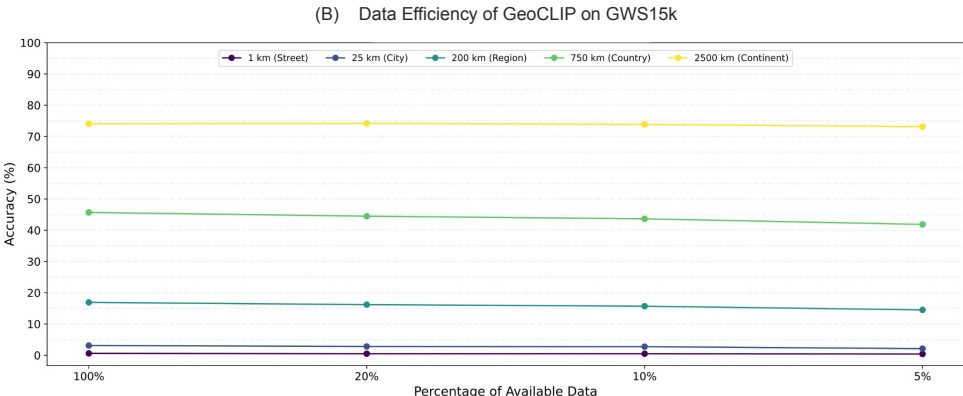

Figure 1: Performance of GeoCLIP in limited data scenarios on (A) YFCC26k dataset and (B) GWS15k datasets. GeoCLIP achieves decent performance across datasets even when the training data is significantly reduced.

We show the efficacy of GeoCLIP on limited training samples of Im2GPS3k in section 4.2 of the main paper. Now, we further investigate the performance of GeoCLIP for limited data settings on other datasets (YFCC26k and GWS15k).

As shown in Figure 1, we observe a similar trend of GeoCLIP on YFCC26k and GWS15k datasets as in the Figure 3 of the main paper. The performance of GeoCLIP is not affected considerably, even when the amount of data is limited. This observation is consistent and holds across datasets. Surprisingly, even if the training data is reduced exponentially (getting as low as 5%), GeoCLIP still achieves competitive performance.

## 3 Additional Ablations

We had discussed a few important ablations (benefits of our GPS encoding and hierarchical learning) in Sec. 4.3 of the paper draft. In this section, we present and discuss other ablations on different components of our method GeoCLIP.

### 3.1 Gallery size

The existing methods, which perform worldwide geo-localization, limit themselves to matching a query image to 21k GPS coordinates (at the finest resolution). They are restricted due to their design choice of predefined classes. However, our method, GeoCLIP, can perform matching against a gallery of any arbitrary length. hence, we vary the gallery size and evaluate GeoCLIP performance on them. The results for the datasets Im2GPS3k [2], YFCC26k [6], and GWS15k [1] are reported in Table 2.

Table 2: Results on the Im2GPS3k [2], YFCC26k [6], and GWS15k [1] datasets when the gallery size is varied.

| Dataset | Gallery Size | Street 1 km | City 25 km | Region 200 km | Country 750 km | Continent 2500 km |
|---|---|---|---|---|---|---|
| **Im2GPS3k [2]** | 21k GPS | 11.88 | 33.10 | 48.75 | 68.70 | 83.18 |
|  | 100k GPS | 14.11 | 34.47 | 50.65 | 69.67 | 83.82 |
|  | 500k GPS | 13.98 | 33.80 | 50.95 | 69.80 | 84.38 |
|  | 1M GPS | 13.98 | 33.47 | 51.48 | 69.14 | 82.85 |
| **YFCC26k [6]** | 21k GPS | 8.44 | 20.28 | 34.52 | 56.37 | 75.16 |
|  | 100k GPS | 11.61 | 22.19 | 36.69 | 57.47 | 76.02 |
|  | 500k GPS | 11.49 | 21.81 | 36.68 | 57.85 | 76.12 |
|  | 1M GPS | 11.45 | 21.51 | 36.57 | 57.68 | 76.05 |
| **GWS15k [1]** | 21k GPS | 0.18 | 2.4 | 14.84 | 42.13 | 73.05 |
|  | 100k GPS | 0.53 | 3.2 | 15.94 | 44.55 | 72.79 |
|  | 500k GPS | 0.6 | 3.12 | 16.92 | 45.69 | 74.06 |
|  | 1M GPS | 0.6 | 2.95 | 16.96 | 46.15 | 74.19 |

The results demonstrate that expanding the GPS gallery improves performance, particularly at smaller scales. For example, on the Im2GPS3k dataset, increasing the gallery size from 21K to 100K leads to an accuracy improvement from 11.88% to 14.11% at the 1km scale. Similarly, augmenting the gallery from 21K to 100K GPS coordinates in the YFCC26k dataset improves the accuracy at the 1km scale from 8.44% to 11.61%. However, the 500K and 1M gallery sizes lead to a slight decrease in performance compared to the 100K gallery. Hence, we used this gallery size to report performance on these datasets.

On the GWS15k dataset, the 500K gallery yields better improvements, increasing the accuracy from 0.18% to 0.6% at the 1km scale. However, further additions to the gallery do not provide more gains. Even though increasing the gallery size to 1M, yields marginally beneficial at larger scales, the performance at smaller scales reduces. Thus, using a 500k gallery size gives a better trade-off, keeping low computational expense and performance on fine-grained scales as a priority.

### 3.2 Queue Length

Negative samples play a crucial role as we use contrastive learning to train our model (refer to eq. 4 of the paper draft). Our method performs image-to-GPS retrieval, thus, the negative samples in our case are the GPS coordinates rather than images used in traditional retrieval methods, which are easy

to obtain. We have shown the benefit of additional negatives via our dynamic queue strategy in Table 2(a) in the main paper. Here, we do further analysis by evaluating the performance when the queue length (S=|Q|) is varied. The results are presented in Table 3.

Table 3: Ablation on the length of the queue used for additional negatives during the training of GeoCLIP.

| Queue Length | Street 1 km | City 25 km | Region 200 km | Country 750 km | Continent 2500 km |
|---|---|---|---|---|---|
| 512 | 7.57 | 32.03 | 48.76 | 67.42 | 82.35 |
| 1024 | 7.79 | 32.63 | 49.22 | 68.39 | 82.68 |
| 2048 | 8.73 | 32.65 | 49.14 | 67.11 | 82.22 |
| 4096 | **9.84** | **33.10** | 49.32 | **69.07** | **83.30** |
| 8192 | 9.64 | 32.00 | 49.01 | 68.12 | 82.82 |
| 16384 | 9.23 | 32.80 | **49.77** | 68.00 | 82.71 |
| 32768 | 8.45 | 32.83 | 49.20 | 68.20 | 82.70 |
| 65536 | 8.31 | 32.48 | 48.59 | 67.52 | 82.19 |

Among the tested queue lengths, $|Q| = 4096$ obtained 9.84%, 33.10%, 49.32%, 69.07%, and 83.30% accuracy on 1km, 25km, 200km, 750km, and 2500km thresholds, respectively. While other queue lengths showed comparable results, the performance of the 4096-length queue is better. Even though the queue length 16384 showed a slight improvement in the 200km threshold, the 4096-length queue still outperformed it on the rest of the thresholds. Hence, we used $|Q|$ as 4096 in our experiments in the main paper.

### 3.3 $\sigma_\eta$ for Batch GPS noise

As explained in section 3.2 of the main paper, we create variation in GPS coordinates by adding a small amount of noise $\eta \sim \mathcal{N}(0, \sigma_\eta^2)$. Here, we perform an ablation where we vary the $\sigma_\eta$ of the Gaussian distribution and report their performance on Im2GPS3k [2] dataset in Table 4. We can observe that $\sigma_\eta$ of 150 meters yields better performance. We used the same value of $\sigma_\eta$ across different datasets for the experiments in the main paper.

Table 4: Ablation on different $\sigma_\eta$ values. We observe the best performance when $\sigma_\eta$ is 150 meters.

| $\sigma_\eta$ (in meters) | Street 1 km | City 25 km | Region 200 km | Country 750 km | Continent 2500 km |
|---|---|---|---|---|---|
| 0 | 8.43 | 32.06 | 47.76 | 65.49 | 81.84 |
| 10 | 8.89 | 32.17 | 47.66 | 67.01 | 81.59 |
| 50 | 8.56 | 31.82 | 47.14 | 66.19 | 81.57 |
| 150 | **8.95** | **32.61** | **47.96** | **67.09** | **82.08** |
| 300 | 8.05 | 31.32 | 46.89 | 65.40 | 81.54 |
| 500 | 7.54 | 31.78 | 47.27 | 65.34 | 81.39 |

### 3.4 $\sigma_{\eta'}$ for Queue GPS noise

As described in the main paper (Sec. 3.2), we propose a dynamic queue strategy with GPS coordinates and use them as additional negative samples n the contrastive learning of GeoCLIP. We also showed the benefit of it in Table 2(a) in the paper draft. We now do further analysis on the queue and perform ablations with respect to the static queue and the gaussian noise parameter $\sigma_{\eta'}$.

As shown in Table 5, compared to the static queue, we observe improvement in performance across all the distance accuracy metrics while using the dynamic queue, especially on the finer scales (1km and 25 km distance thresholds). Note that the performances reported are without hierarchical learning. Moreover, adding noise sampled from gaussian distribution ($\eta' \sim \mathcal{N}(0, \sigma_{\eta'}^2)$) to the GPS coordinates of the dynamic queue leads to further gains in performance. $\sigma_{\eta'}$ of 1000 meters yields improvements on all distance thresholds compared to dynamic queue performance, while for $\sigma_{\eta'} > 1000m$, the performance on smaller scales improved at the cost of drop in accuracy on the large scales.

Table 5: Comparison of the performance of our dynamic queue strategy with the static queue. We also perform an ablation on $\sigma_{\eta'}$ used to add gaussian noise to the dynamic queue. The addition of noise to GPS coordinates of dynamic queue yields further gains in performance.

| Method | Street 1 km | City 25 km | Region 200 km | Country 750 km | Continent 2500 km |
|---|---|---|---|---|---|
| Static Queue | 5.93 | 31.8 | 49.02 | 67.42 | 82.22 |
| Dynamic Queue | 7.55 | 32.91 | 49.28 | 67.46 | 82.48 |
| Dynamic Queue + noise ($\sigma_{\eta'} = 1000$m) | 9.84 | 33.10 | 49.32 | 69.07 | 83.35 |
| Dynamic Queue + noise ($\sigma_{\eta'} = 5000$m) | 11.85 | 33.07 | 48.90 | 67.60 | 82.87 |
| Dynamic Queue + noise ($\sigma_{\eta'} = 10000$m) | 11.53 | 33.51 | 48.63 | 67.55 | 82.66 |
| Dynamic Queue + noise ($\sigma_{\eta'} = 25000$m) | 10.01 | 32.85 | 47.76 | 67.22 | 82.46 |

## 3.5 $\sigma$ for Random Fourier Features

We employed positional encoding with Random Fourier features (RFF) in our location encoder (discussed in Sec. 3.1.1) to overcome the spectral bias (favoring low-frequency) of direct usage of MLPs. In RFF, the $\sigma$ value plays an important role in determining the range of frequencies. In an ideal scenario where a single metric is being optimized, a specific optimal sigma value could be determined through a hyperparameter search. However, in our task, we evaluate our method on five different distance metrics. Based on our extensive experiments when searching for an optimal $\sigma$ value on a particular metric, we found a particular pattern.

In Figure 2, we train our GeoCLIP model by varying the $\sigma$ value on a wide range (from $2^0$ to $2^{12}$), and evaluate their performance on different threshold metrics. We observe an interesting trend. A single $\sigma$ value does not perform best for all the metrics. We observe that higher $\sigma$ values are preferable for fine-grained scales (1km and 25km). Conversely, for coarser scales (200km, 750km, and 2500km), lower $\sigma$ values perform better than their higher $\sigma$ counterparts.

Based on the above observations, a simple approach would be to select intermediate $\sigma$ values to achieve reasonably good performance across multiple scales. Hence, we showed results of our method on the intermediate $\sigma$ value in Table 2 (a) in the main paper. However, this strategy is suboptimal. Our proposed hierarchical strategy outperforms the single $\sigma$ value approach when optimizing metrics at different scales (Table 2 (b) in the main paper), emphasizing the combination of hierarchies being better than their individual parts.

## 3.6 Number of hierarchies ($M$)

In Table 2(b) of the main paper, we demonstrate the importance of hierarchical learning to perform well across all the distance metrics. In Table 6, we further conduct experiments with varying numbers of hierarchies ($M$) to identify the optimal $M$ that maximizes performance across all metrics. We observe that increasing the number of hierarchies leads to improved model performance up to three hierarchies. However, beyond that, increasing the hierarchies leads to a slight decrease in performance. Hence, in the main paper, we utilized three hierarchies in our GeoCLIP model.

Table 6: Performance of GeoCLIP when the number of hierarchies is varied.

| Number of hierarchies | Street 1 km | City 25 km | Region 200 km | Country 750 km | Continent 2500 km |
|---|---|---|---|---|---|
| 1 | 9.84 | 33.1 | 49.28 | 69.07 | 83.35 |
| 2 | 13.85 | 32.9 | 49.3 | 69.3 | 83.52 |
| 3 | **14.11** | **34.47** | **50.65** | **69.67** | **83.82** |
| 4 | 14.01 | 34.33 | 50.52 | 69.54 | 83.22 |
| 5 | 13.98 | 34.48 | 49.88 | 68.9 | 82.85 |

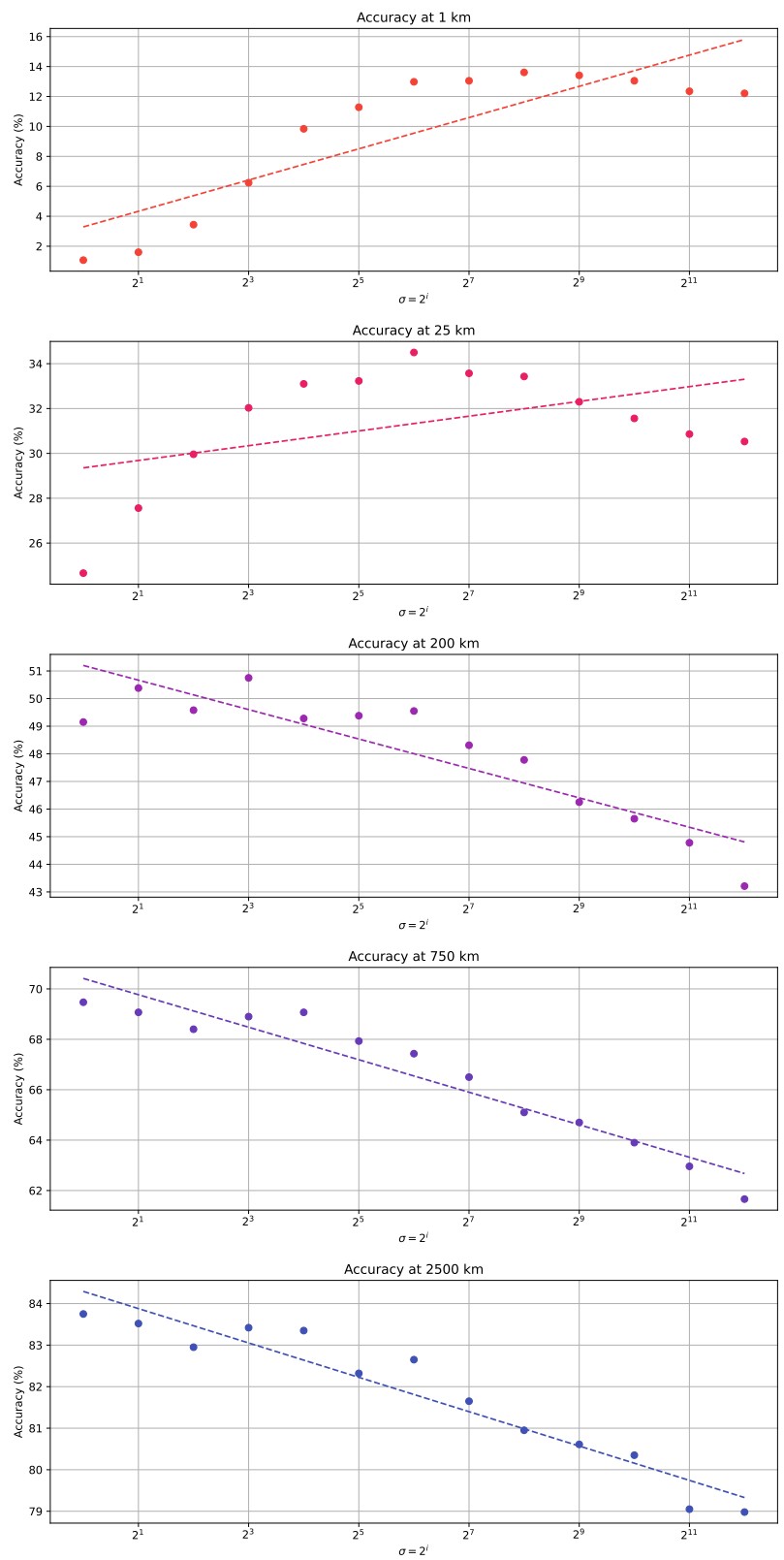

Figure 2: Ablation on different $\sigma$ values (single hierarchy) used to generate the Random Fourier Features. We train our model by varying $\sigma$ from $2^0$ to $2^{12}$ and measure the performance across different distance thresholds. The trend is shown by dashed lines. Higher $\sigma$ values are preferred on smaller scales (1km and 25km) while larger scales (200km, 750km and 2500km) performs well with smaller $\sigma$ values.

# 4 Different selection choices for GPS Gallery Construction

The primary approach for constructing the GPS gallery in our experiments in the main paper involves sampling a predetermined number of coordinates from the training dataset. This strategy aligns with prior geo-localization methods that also leverage the training set information to create discrete geographical cells (classes), while we use the GPS coordinates information of the training set to construct the reference GPS gallery. The underlying rationale is that the training set is likely to contain locations of interest, defined as places where photographs have been taken before and are thus probable sites for future image captures. Given an image with an unknown location, these areas serve as initial points of reference, making this method particularly relevant for real-world evaluation scenarios. However, this approach operates under specific assumptions about the geographical distribution of future test data. To explore GeoCLIP's performance without such assumptions, we also investigate alternative choices for gallery construction, as explained below.

## 4.1 Evenly Spaced GPS Coordinates

To probe GeoCLIP's adaptability, we construct GPS galleries using evenly spaced coordinates across the Earth's surface. This approach provides a test of GeoCLIP's generalization capabilities, particularly when the geographical distribution of the images is unknown or not confined to specific regions. Under this scenario, we adopt two distinct approaches for gallery construction:

**Global Sampling:** In this scenario, we generate a gallery containing 1 million coordinates that are evenly distributed across the entire sphere of the Earth. This includes both land and oceanic regions. The distribution is achieved using a Fibonacci lattice, ensuring an even spread of coordinates. In this case, we do not assume any prior geographical knowledge about where the images might be located.

**Land-Only Sampling:** In this case, while similar to the previous, we focus exclusively on terrestrial regions. Here, the gallery comprises 1 million coordinates, also arranged using a Fibonacci lattice but limited to land areas. This scenario reflects situations where we have the additional information that the image of interest is taken on land, thus reducing the search space.

By employing these alternative GPS galleries, we aim to assess GeoCLIP's performance in diverse geo-localization contexts. The results for both the scenarios are presented below:

Table 7: Performance of GeoCLIP when the gallery is constructed by evenly sampling GPS coordinates.

| GPS Gallery | Street 1 km | City 25 km | Region 200 km | Country 750 km | Continent 2500 km |
|---|---|---|---|---|---|
| Evenly Spaced (1M Land Only) | 0.02 | 12.85 | 37.07 | 59.39 | 77.78 |
| Evenly Spaced (1M All Earth) | 0.03 | 9.18 | 33.47 | 55.32 | 75.34 |

Without using any dataset prior knowledge of geographical places of interest, the reference gallery can be constructed by evenly sampling GPS coordinates across the globe, as shown above. But, it is not a better choice to create a GPS gallery as there can be many places on Earth (like oceans and polar regions), which may not be relevant places of interest (also evidenced by results). To accurately predict within a 1km radius, GPS coordinates must lie in the reference gallery. By uniformly sampling 1 million coordinates across the Earth's surface, a total area of $1,000,000$ km² is encompassed. However, considering Earth's land area of $148,326,000$ km², the likelihood of having the true coordinates in the reference gallery is low. Thus, if true GPS coordinates fall outside the available GPS options of the gallery (for instance, not within a 1km proximity to the true location), then image-to-GPS performance within a 1km distance threshold would suffer. As we observe in higher scales, competitive performance can be achieved through uniform sampling, given the increased likelihood of true coordinates lying within the larger distance thresholds from reference gallery coordinates. Hence, given the vast search space, applying prior knowledge and alternative search methods can enhance efficiency and improve performance.

## 4.2 Test Set GPS Coordinates

Similar to traditional image-to-image retrieval methods, the image gallery can also be constructed using the test set (in our case, test set GPS coordinates). As shown in Table 8, we obtain strong localization performance across different distance threshold metrics when a GPS gallery is constructed

using the test set. However, such an approach may lack generalizability, as it essentially reduces the problem to a retrieval task among known, labeled locations.

Table 8: Performance of GeoCLIP when the gallery is constructed test set GPS coordinates.

| Dataset | Street 1 km | City 25 km | Region 200 km | Country 750 km | Continent 2500 km |
|---------|-------------|------------|---------------|----------------|-------------------|
| Im2GPS3k | 29.66 | 47.64 | 59.05 | 72.53 | 85.78 |

These alternative methods for GPS gallery construction, as discussed in Sec. 4.1 and 4.2, offer deeper insights into GeoCLIP's robustness and versatility across varying geographical scales and conditions.

## 5 Analysis of Runtime and Memory Footprint

To provide a comprehensive understanding of GeoCLIP's efficiency, we compare its runtime and memory footprint with baseline methods, specifically ISNs [4] and Translocator [5]. The table 9 summarizes the findings.

Table 9: Comparison of Runtime and Memory Footprint

| Methods | Memory Footprint (parameter count) | Runtime (batch of 128 images) |
|---------|-----------------------------------|-------------------------------|
| ISNs [4] | 257,830,177 | 0.0997 |
| Translocator [5] | 478,125,882 | 0.2374 |
| GeoCLIP (Ours) | 314,400,770 | 0.1043 |

Even though GeoCLIP's memory footprint and runtime are higher than those of ISNs [4], they are substantially lower than Translocator [5]. Importantly, GeoCLIP outperforms both methods across all distance threshold metrics, indicating that the trade-off between efficiency and effectiveness is favorable.

It's worth noting that classification-based methods like ISNs generally have lower runtimes as they do not involve searching through a gallery of images. On the other hand, methods like Translocator, which incorporate auxiliary information such as scene contexts via a dual-branch vision transformer architecture, tend to have higher runtime and parameter counts [5]. In contrast, GeoCLIP achieves better performance without relying on additional scene-based information and employs a lightweight location encoder composed of MLPs with four hidden layers and one output layer.

## 6 Motivations for using Pretrained CLIP as Image encoder Backbone

In our proposed method, GeoCLIP, we employ a CLIP backbone as part of our image encoder. We leverage the CLIP model as it has been originally trained on 400 million (image, text) pairs and has shown strong generalization performance across different downstream tasks. We keep the CLIP backbone frozen in our method as our design choice is influenced by computational constraints and the nature of our training dataset. The MP-16 dataset [3], a standard for worldwide geo-localization, contains $4.72$ million geotagged images. Training from scratch or fine-tuning the entire image encoder on such an extensive dataset demands high computational resources and considerable training time. These factors present challenges, particularly in academic settings with limited resources. To mitigate these computational costs, we freeze the CLIP backbone and only train the additional linear layers ($h_1$ and $h_2$) introduced to adapt the model to the geo-localization task.

Another benefit of using pretrained CLIP weights is that it allows precomputation of CLIP features, further reducing computational overhead. We observed precomputing training set features drastically reduced our training time. For instance, the training time of one epoch on the MP-16 dataset was reduced substantially from $\approx 27$ hrs to $\approx 15$ minutes.

Interestingly, our empirical evaluations corroborate the efficacy of employing a frozen CLIP backbone. In contrast, fine-tuning the CLIP backbone did not improve performance; in fact, the frozen backbone performed better. This may be the consequence of catastrophic forgetting of original CLIP weights while training. Additionally, the use of a frozen and publicly accessible CLIP model has ethical implications. Given that CLIP is a publicly available model, it enables the implementation of defense mechanisms, such as adversarial noise, which allows individuals to protect their privacy better.

Moreover, CLIP is a multimodal architecture (its image and text encoders are aligned). When we align the pretrained image encoder of CLIP with our location encoder, the pretrained text encoder of CLIP also gets implicitly aligned with the location encoder. As a result, without separate training on text, our method, GeoCLIP, can also process text queries, enabling geo-localization using text.

# 7 Qualitative Demonstration

In this section, we qualitatively demonstrate the effectiveness of our method. We first visually demonstrate the geo-localization by different hierarchies of our location encoder. In the subsequent subsections, we show the results of GeoCLIP on both the image query and text query. Finally, we provide a visualization of the distribution of correct predictions of our model and compare them with the actual test dataset distributions.

## 7.1 Hierarchical learning in our location encoder $\mathscr{L}(\cdot)$

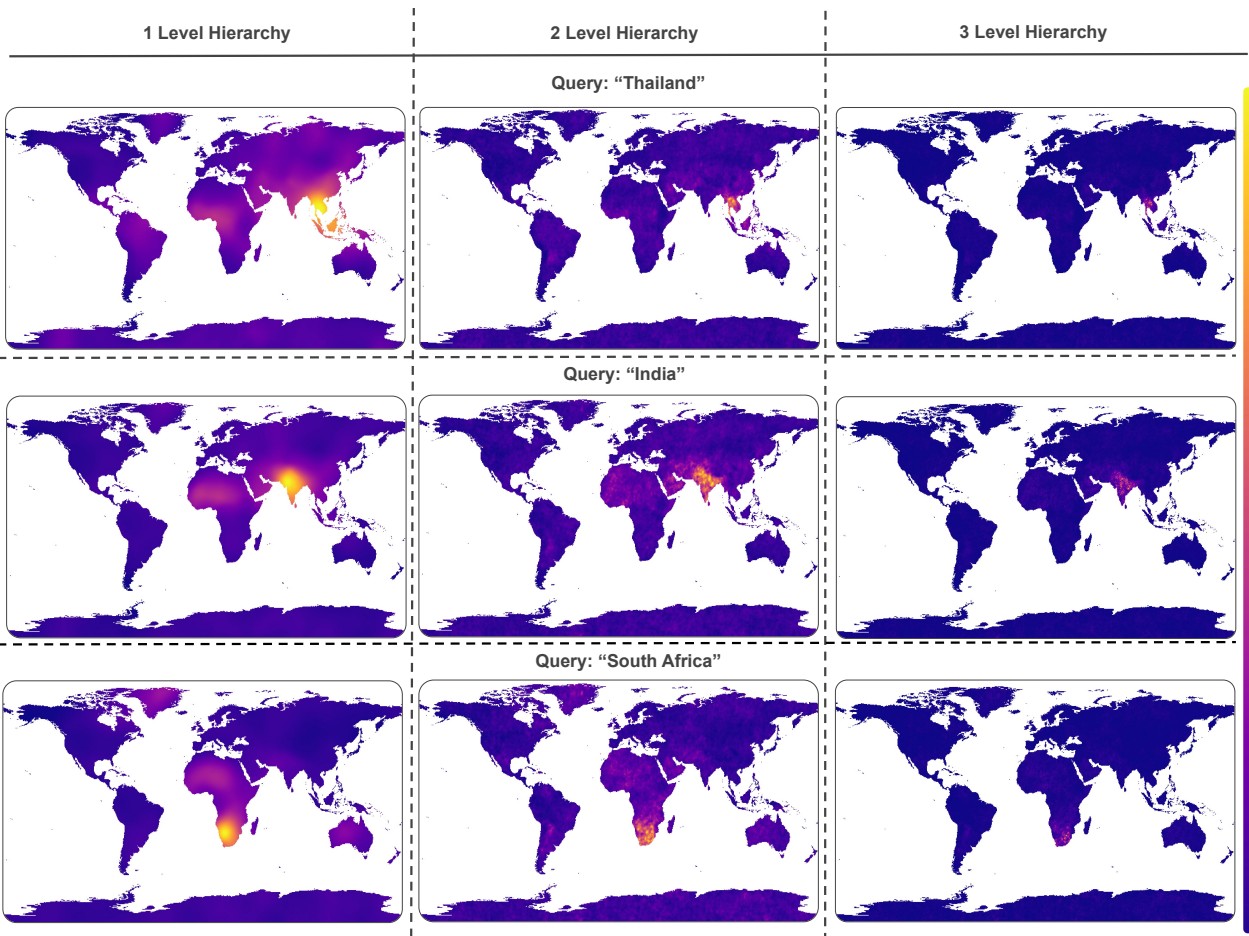

Figure 3: Visualization of the geo-localization of different places mentioned in the query. For each query, we show the heatmap of predictions using 1-level, 2-level, and 3-level hierarchies. The 1-level hierarchy performs localization at a coarse resolution while increasing the hierarchies helps in localizing more precisely.

## 7.2 GeoCLIP with Image Query

| Error $\leqq$ | Images |
|---|---|
| **1km** | 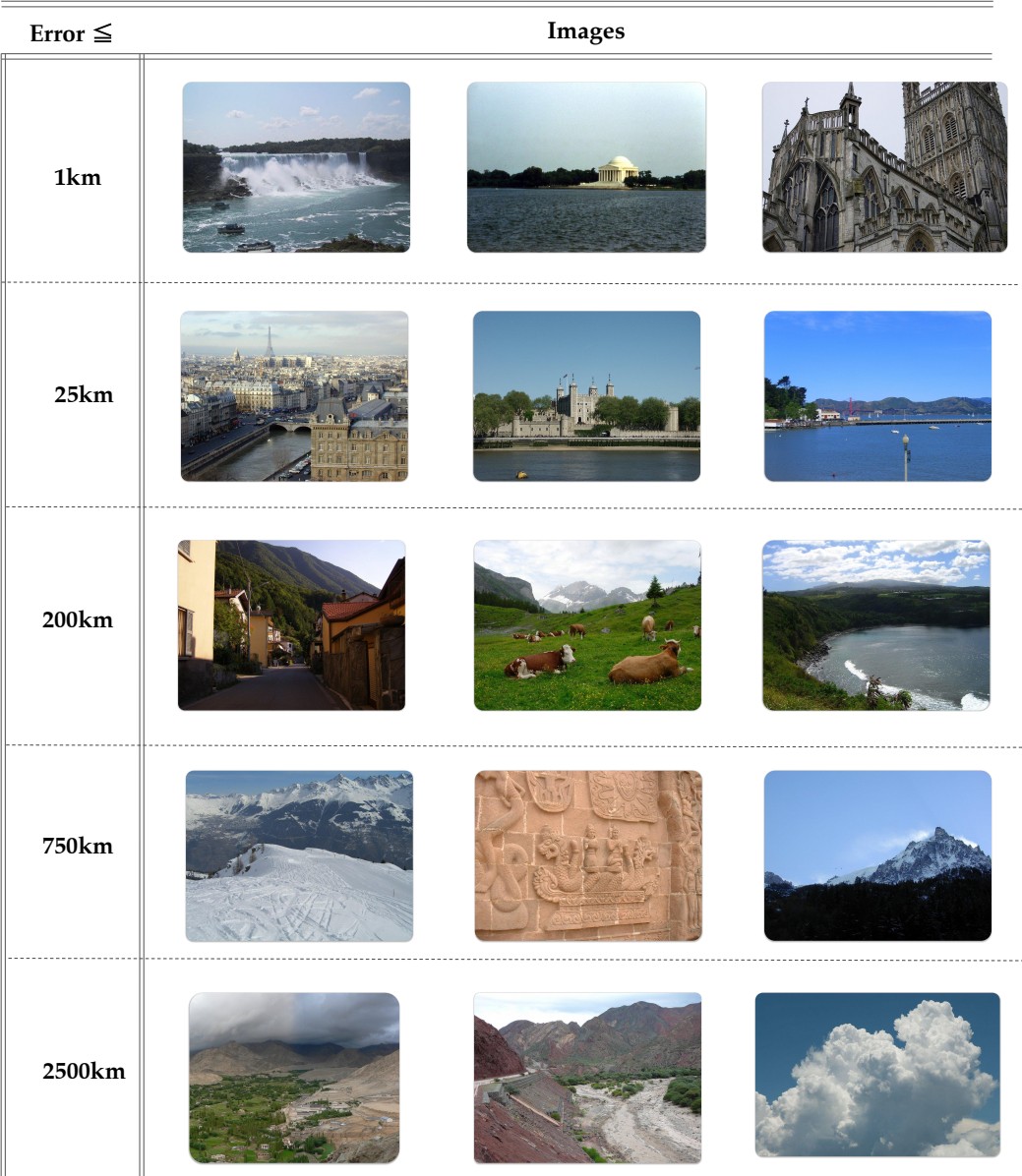 |
| **25km** | |
| **200km** | |
| **750km** | |
| **2500km** | |

Figure 4: Sample query images from Im2GPS3k dataset [2] on which GeoCLIP localizes within an error of 1km (1st row), 25km (2nd row), 200km (3rd row), 750km (4th row) and 2500km (last row).

## 7.3 Distribution of correct predictions of GeoCLIP on different datasets

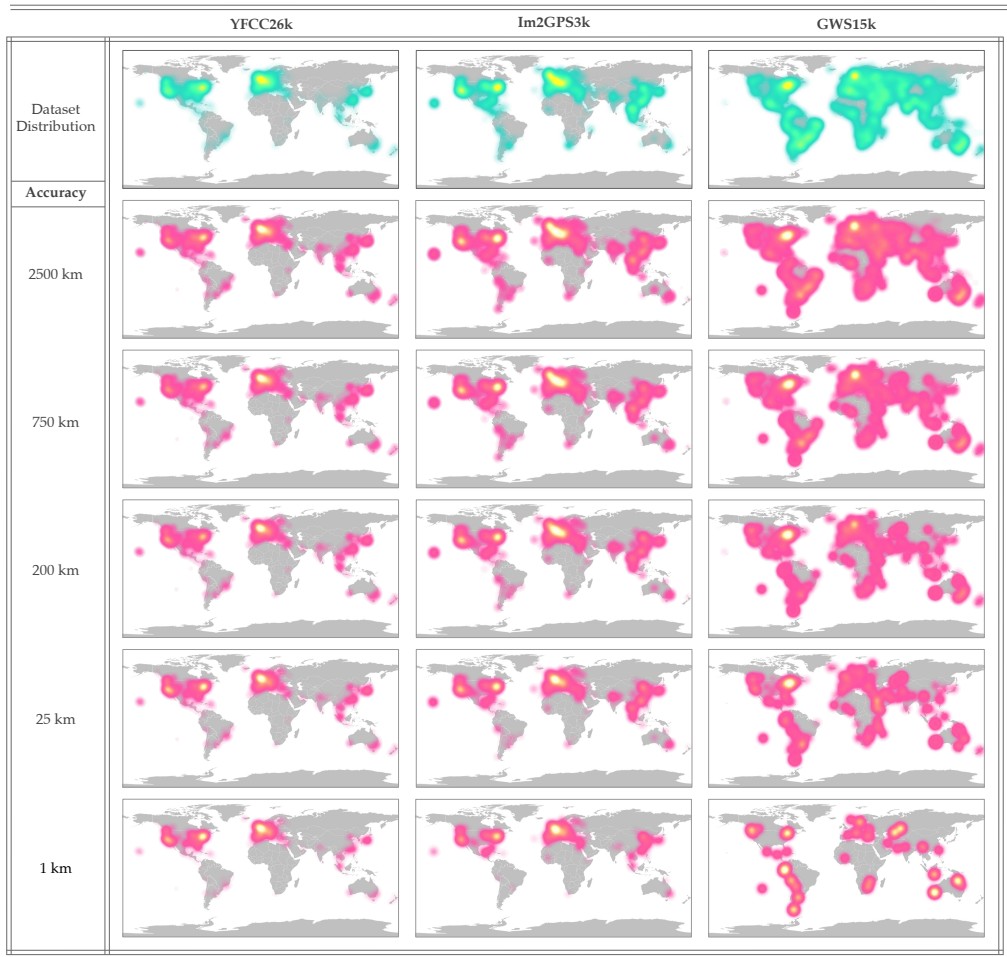

Figure 5: Demonstration of the distribution of correct predictions of GeoCLIP at different threshold metrics on various benchmark datasets. Our predictions match closely with the dataset distribution.

## 7.4 GeoCLIP with Text Query

**Specific Landmarks and Places:**

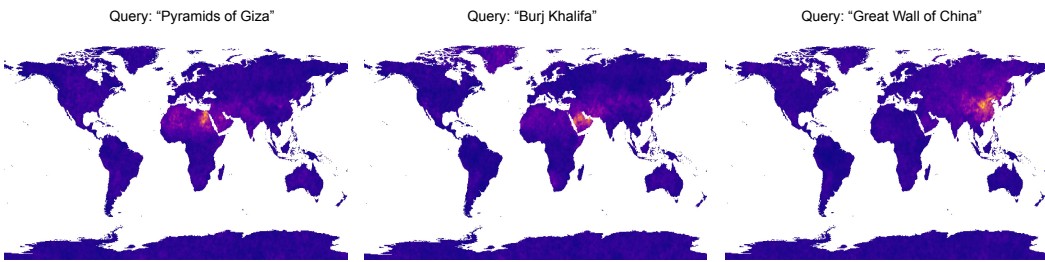

Figure 6: Geo-localization using GeoCLIP with text query as specific landmarks.

(a) **Text Query: "Boston, Massachusetts"**

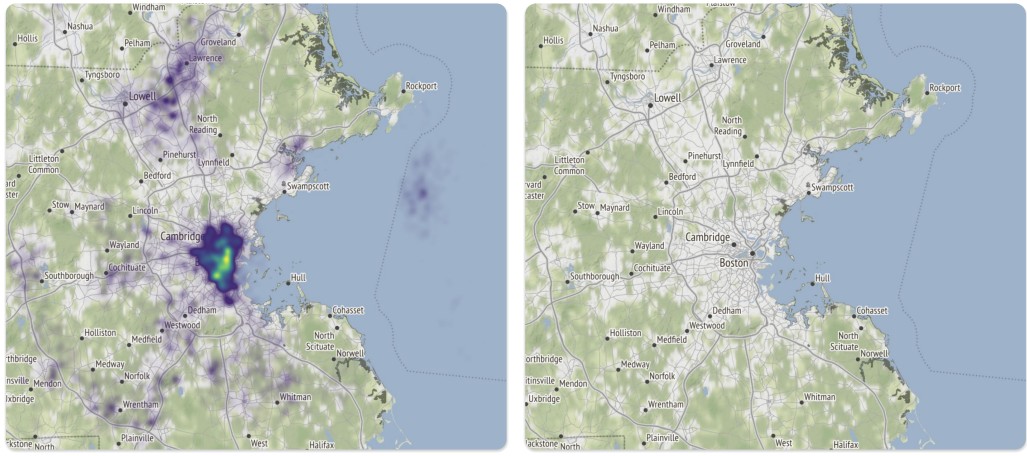

(b) **Text Query: "Colosseum, Rome"**

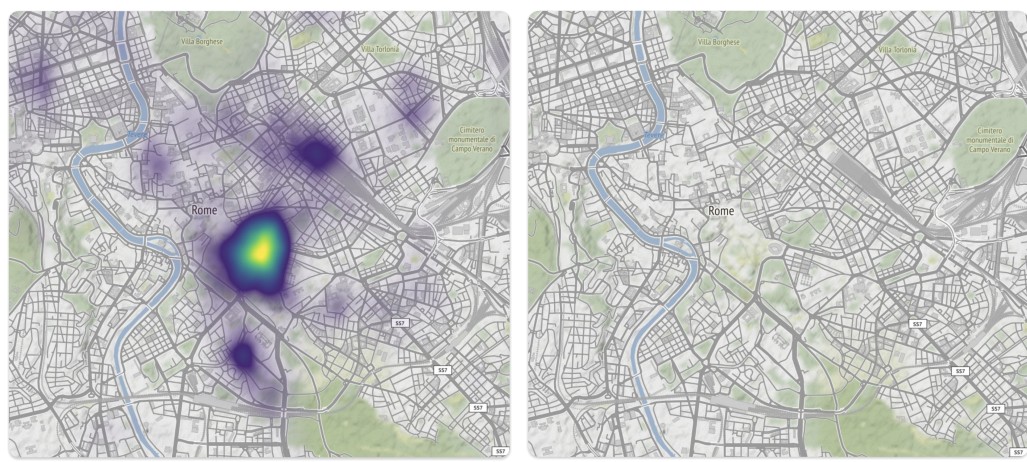

(c) **Text Query: "Niagara Falls"**

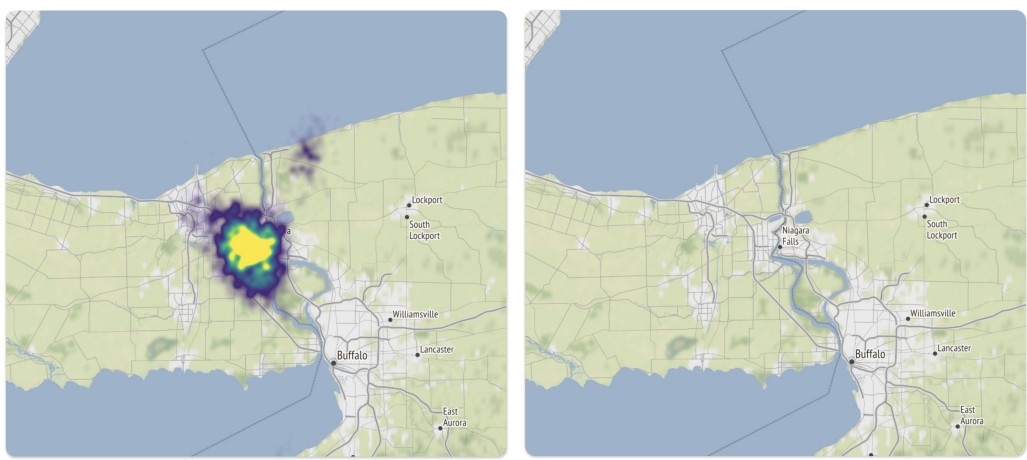

Figure 7: Across different text queries ranging from city to specific places, we demonstrate the geo-localization using our method GeoCLIP on the left, while the original region is shown on the right.

# 8 Discussion on Ethical Issues and Possible Mitigation

The capability of our GeoCLIP model to accurately geo-localize images on a global scale introduces ethical considerations related to privacy and security. Individuals may be apprehensive about the potential misuse of such technology, especially when it comes to revealing their geographic location without consent.

In response to these concerns, we have designed GeoCLIP with countermeasures in mind. Our method employs a publicly accessible, pre-trained CLIP backbone for the image encoder, allowing individuals to directly implement defense mechanisms. Specifically, a form of defense can be achieved through the addition of adversarial noise to images. This carefully crafted human-imperceptible noise can significantly alter the image embeddings generated by the CLIP backbone, thereby disrupting the model's ability to accurately predict geographical locations. For instance, the adversarial noise can be tailored to maximize the image embedding for an unrelated target prompt, such as "Eiffel Tower," leading to incorrect matches with the GPS embeddings in the gallery. This approach leverages the public availability of the pre-trained CLIP model weights, offering individuals a viable means to protect their location privacy. Thus, GeoCLIP aims to strike a balance between technological utility and ethical considerations.