# OpenReview forum: "GeoCLIP: Clip-Inspired Alignment between Locations and Images for Effective Worldwide Geo-localization"
_NeurIPS.cc/2023/Conference — NeurIPS 2023 poster_

### Official Review · Reviewer_SMLq · 2023-06-29

**Soundness:** 4 excellent
**Presentation:** 3 good
**Contribution:** 3 good
**Rating:** 6
**Confidence:** 4

**Summary:**

This paper tackles the image geolocalization problem. Instead of formulating the task as image retrieval or classification over a set of geospatial bins as is typical, the authors frame it as image-to-GPS retrieval by matching image features with location features. The proposed training strategy leverages Random Fourier Features (RFF) to encode geospatial location across multiple scales and align them with features from a CLIP visual encoder, with several training enhancements. The resulting method achieves state of the art performance on three benchmarks, across all scales, and showcases interesting applications in text geolocalization and classification.

**Strengths:**

- Addresses an important and interesting problem. Image localization is a fundamental problem in vision that continues to receive attention in the literature.

- The approach has technical novelty and is of the first to formulate the localization task as an image-to-GPS retrieval problem. The key idea is to relate images directly to locations via aligning image features to location embeddings using contrastive learning. Such an approach uses a continuous model of location whereas traditional geolocalization approaches depend on a large reference database of images with known locations to support image-level matching (via image retrieval or classification) and inferring location from the closest match. Instead, the authors propose the concept of a GPS gallery.  Though training still requires geotagged images (image + location), the authors show the proposed approach requires less training data.  Further, at inference, the GPS gallery can theoretically be any set of locations to query against, instead of being limited by the reference image set. Geospatial location is represented using NeRF-style location embeddings (Random Fourier Features with an MLP).  Results show that infusing the location representations with multiple spatial scales improves geolocalization performance.

- Evaluation is extensive and compelling. State-of-the-art (SOTA) performance on multiple datasets at different threshold scales (street, city, region, country, and continent). Near SOTA performance on Im2GPS3k training using only 20% of the training data.

- Intriguing applications. The results on text geolocalization and geotagged image classification showcase the utility of the location encoding beyond image-based geolocalization.

**Weaknesses:**

- A central idea of this paper is the concept of modeling location continuously. I was disappointed to see that during evaluation, the GPS gallery (the set of possible locations) was limited to locations sampled from the training set (L230). Though I understand why this was done from an evaluation perspective, it significantly hurts the "global localization" theme. At the very least, I would have liked to see an evaluation of how selecting the GPS gallery at inference (choice of sampling distribution, granularity, etc.) impacts performance. This also raises the issue that the location encoder is not truly global, it is limited by where images have been captured on the globe (as training requires image/location pairs).

- Some minor details missing. Section 4.2 does not mention which dataset the experiment was performed on. Comparing the numbers from Figure 3 and Table 1a, it can be assumed to have been conducted on Im2GPS3k, but the dataset should still be mentioned to improve clarity. Similarly, section 4.3 does not mention which dataset(s) was/were used for the ablation study.

- The application for text query-based geolocalization is interesting, but the results are lacking. It is only demonstrated qualitatively on a continent scale using a single keyword (desert) in the main paper. While this is interesting, it leaves a lot to be desired in terms of understanding the full or even partial capability of the text encoding for tasks such as geolocalization or digital forensics.

- Quality of the presentation could be improved. For example, the introduction could provide additional motivation for the underlying problem. In its current form, the introduction mentions that there are a variety of applications for geolocalization, but does not support this claim with any citations or descriptions of how geolocalization benefits them. Instead, the majority of content is related to why the problem is difficult and how existing approaches are inadequate, repeating much of the related work section content (e.g., binning, classification, and scaling methods). The conclusion section, in particular, lacks flow and could benefit from additional editing.  Unclear mathematical notation on L138. What is K?  Minor grammatical errors (L49, L107, L111, L372, etc.).

- Word choice in the conclusion. L353 suggests that this work “solved worldwide geolocalization”. Of course this problem is far from solved!

**Questions:**

My initial rating is weak accept. This paper introduces a method for training a high-quality geospatial location encoder that interacts with, widely used, CLIP features and demonstrates its use for localization and several applications. This has the potential to benefit the wider community.

Quantitatively, how does the proposed approach compare in terms of runtime and memory footprint with existing approaches?

How does the text encoding perform at smaller scales (region or city)?

Can the text encoding benefit the image-based geolocalization?

**Limitations:**

Limitations addressed in the manuscript.

---

> ### Author Rebuttal · Authors · 2023-08-09
>
> We thank the reviewer for acknowledging our problem as “important and interesting”, for finding our approach and our problem formulation to be “novel”, for finding our evaluation to be “extensive and compelling”, and for mentioning our demonstrated applications as “intriguing”. We reply to the queries below.
>
> **Results on different selection choices for GPS gallery**: As suggested, we perform evaluations when the GPS gallery is formed by different sampling distributions and present the results on Im2GPS3k dataset below.
>
> | GPS Gallery                        |  1km  |  25km | 200km | 750km | 2500km |
> |------------------------------------|:-----:|:-----:|:-----:|:-----:|:------:|
> | Sampling from Train Set (lat, lon) | 14.11 | 34.47 | 50.65 | 69.67 |  83.82 |
> | Test set (lat, lon)                | 29.66 | 47.64 | 59.05 | 72.53 |  85.78 |
> | Evenly Spaced (1M Land Only )      |  0.02 | 12.85 | 37.07 | 59.39 |  77.78 |
> | Evenly Spaced (1M All Earth)       |  0.03 |  9.18 | 33.47 | 55.32 |  75.34 |
>
> Similar to traditional image-to-image retrieval methods, when the image gallery is constructed using the test set (in our case, test set GPS coordinates), we achieve the best performance. To have a fair comparison with existing worldwide geo-localization methods (classification-based approaches), which use training set to create geographical cells, we used the GPS coordinates of the training dataset to construct the reference GPS gallery.
>
> Without using any dataset prior knowledge of geographical places of interest, reference gallery can also be constructed by evenly sampling GPS coordinates across the globe. But, it is not a good way to create GPS gallery as there can be many places on Earth (like oceans and polar regions), which may not be relevant places of interest (also evidenced by results). To accurately predict within 1km radius, GPS coordinates must lie in the reference gallery.  By uniformly sampling one million coordinates across Earth's surface, a total area of 1,000,000 km² is encompassed. But, considering Earth's land area $\approx$ 148,326,000 km², the likelihood of having the true coordinates in reference gallery is low. Thus, if true GPS coordinates fall outside the available GPS options of the gallery (for instance, not within a 1km proximity to the true location), then image-to-GPS performance within 1km distance threshold would suffer. As we observe in higher scales, competitive performance may be achieved through uniform sampling, given the increased likelihood of true coordinates lying within the larger distance thresholds from reference gallery coordinates. Given the vast search space, applying prior knowledge and alternative search methods can enhance efficiency and improve performance.
>
> We also perform evaluations when granularity of sampling is varied and show the results below.
>
> | Granularity |  1km  |  25km | 200km | 750km | 2500km |
> |:-----------:|:-----:|:-----:|:-----:|:-----:|:------:|
> |  Continent  | 15.02 | 37.70 | 55.39 | 78.11 |  100.0 |
> |   Country   | 15.85 | 41.18 | 63.33 | 100.0 |  100.0 |
> |    State    | 18.82 | 52.12 | 100.0 | 100.0 |  100.0 |
> |     City    | 26.15 | 100.0 | 100.0 | 100.0 |  100.0 |
>
> We considered four different granularity levels. The performance of image-to-GPS retrieval improves as we search from coarse to finer granularity, which is quite intuitive. For instance, if we have prior knowledge of the city, it's much easier to geo-localize the true location than to know the continent.
>
> **Missing minor details**: Thank you for pointing this out. Yes, results in sec. 4.2 and 4.3 are on Im2GPS3k dataset. We will mention it in the final version to improve clarity and avoid confusion.
>
>  **Mathematical notation and minor grammatical errors**: K is the total number of query images in $D_{test}$. We will mention it and correct the grammatical errors as suggested.
>
> **Word choice in conclusion**: We will edit that line as suggested.
>
> **Comparison of runtime and memory footprint**: Below we provide runtime and memory footprint of baseline methods (ISNs, Translocator) and compare our method.
>
> | Methods           | Memory footprint (parameter count) | Runtime (batch of 128 images) |
> |-------------------|:----------------------------------:|:-----------------------------:|
> | ISNs [12]         |             257,830,177            |             0.0997            |
> | Translocator [14] |             478,125,882            |             0.2374            |
> | GeoCLIP (Ours)    |             314,400,770            |             0.1043            |
>
> Though our method has higher storage and runtime than ISNs, is much lower than Translocator. Note that, GeoCLIP has better accuracy than both approaches across all distance threshold metrics. In general, classification-based methods are expected to have lower runtime as no searching is performed on gallery. But, recent methods such as Translocator use auxiliary information such as scenes to get better performance by formulating a multi-task learning framework using dual branch vision transformers, which can lead to higher runtime apart from higher parameters. Our method does not depend on additional information such as scenes and has a lightweight location encoder containing MLPs of 4 hidden layers and one output layer.
>
> **Performance of text encoders at smaller scales (region or city)**: We qualitatively demonstrate the localization of a city and landmarks in the figure of attached pdf. Please note GeoCLIP using text queries on a few specific places is also provided in supplementary section 4.3.
>
> **Text encoding to benefit image-based geolocalization**: Different modality information, such as text, can benefit image-based geolocalization. One simple way could be to fuse text and image features, then use resultant features for matching GPS embeddings. However, our current training dataset of geo-localization does not have (image, text) pairs. But this idea has great potential and can be an interesting future work.

---

> > ### Comment · Reviewer_SMLq · 2023-08-14
> >
> > Thanks for your response. I have read the other reviews and author rebuttals. All reviewers are in agreement and I will be retaining my initial rating. I think the information provided here (e.g., the strategy for constructing the GPS gallery) and across the other responses is important and I encourage the authors to include these results/discussion points as updates to the supplemental material (or in the main document when appropriate).

---

> > > ### Author Response · Authors · 2023-08-15
> > >
> > > We sincerely thank the reviewer for thoroughly reviewing our rebuttal responses and for retaining the initial rating. As suggested, we will appropriately include the results and other important discussions provided in the rebuttal in the final version of the manuscript, either in the main paper or in the supplementary.

---

### Official Review · Reviewer_AAT4 · 2023-07-04

**Soundness:** 4 excellent
**Presentation:** 3 good
**Contribution:** 3 good
**Rating:** 6
**Confidence:** 4

**Summary:**

The paper addresses the problem of geo-locating an image, i.e. estimating the latitude and longitude of the where the image was taken.

Previous works treat the problem as a classification problem (of cells distributed across the globe). In contrast the this work proposes to retrieve the GPS coordinates from a collection of possible locations. In order to do so the authors propose a neural network architecture that encodes the (ground truth) GPS location and the image in a joint feature space and train it via a contrastive learning scheme. During the inference the query image is encoded and the GPS coordinate closest in feature space is taken as the predicted location.

The main contribution of the paper is the positional encoding of GPS coordinates via random fourier features [21], which are demonstrated to help represent higher frequency information and hence improve localization at higher spatial resolution (Tab. 2).

Experimental results demonstrate state-of-the-art retrieval performance on 3 different datasets.

**Strengths:**

**S1** The encoding of GPS coordinates via random fourier features (RFF) is novel. While RFF have been demonstrated to be useful to capture higher frequency properties for 2d image or 3d coordinates [21], e.g. in NeRF, to the best of my knowledge they have not been used to encode lat/lng coordinates for geo-localization. The authors also demonstrate that an equal earth projection (EEP) (which preserves the area covered) is beneficial for encoding 2d coordinates on the globe, especially for higher accuracy localization (< 1km in Tab. 2).

**S2** Good ablation studies that demonstrate the usefulness of the chosen solutions, especially Tab. 2 for EEP and RFF, but also the ablations provided in the sup. material in Sec. 3. This helps the reproducibility of the approach.

**S3** The place recognition performance improves considerably over state-of-the-art, even though the taken approach is relatively simple. This can be attributed to the ability of the model to learn an understand of spatial information.

**Weaknesses:**

**W1** The usage of the CLIP frozen backbone is not well motivated and evaluated. As such it remains unclear why CLIP features are needed for the place recognition task, rather than a network trained from scratch or fine-tuned on the task. Experimental evaluation demonstrates how the learned feature is able to relate textual information to geographic locations, but the influence of this capability on predicting lat/lng locations for a query image is not evaluated.

**W2** The evaluation of the text-to-location encoding in Sec. 4.5 is very limited. The encoded location is used for image classification, but the paper falls too short in outlining the goal and metric of the chosen classification task. As such the evaluation can not stand by it's own but the reader needs to resort to [4, 31] for context.

**W3** The size of the gallery (the quantization of possible GPS locations) is tuned per dataset. Instead a single size across all dataset would foster broader applicability of the approach. Sec. 3.1 of the sup. material suggests that this would in fact be possible, e.g. with a size of ~250k. (See also Q2)

**Questions:**

**Q1** We do you keep the CLIP backbone frozen?

**Q2** Why is the gallery of GPS coordinates sampled from the *training* set? Is this on the dataset from [9] which is used for training? Did you consider covering the globe, e.g. uniformly?
The dependency of the geographic locations on the training set implies that the test set needs to be covered by the training set. This seems to prevent correct out-of-distribution predictions.

**Limitations:**

The ability to accurately locate any photo on the earth represents a security and privacy issue. E.g., even if a user ops out of GPS tagging their photos, the presented approach enables to infer the user location from their images -- and the location trace of the user over time.
This surveillance capability should be stated as negative societal impact or potential harmful consequence of the approach, when deployed in public.

---

> ### Author Rebuttal · Authors · 2023-08-08
>
> We thank the reviewer for acknowledging our GPS encoding method (“encoding GPS coordinates via random fourier features”) to be “novel”, for mentioning our representation of 2D GPS coordinates using equal earth projection to be “beneficial”, for finding our ablation studies to be “useful”, and for appreciating our ablations in Table 2 and in the supplementary material (Sec. 3). We reply to the queries below.
>
> **Frozen CLIP backbone**: We use CLIP backbone in our image encoder, which is kept frozen during training. We also add two linear layers (h1 and h2) to adapt the backbone to our geo-localization task, where these layers are kept trainable (Lines 153-157). The reasons for keeping the CLIP backbone frozen are as follows:
> * The standard training dataset of worldwide geo-localization (MP-16) contains 4.72 million geotagged images (Lines 237-238). Training from scratch or finetuning the whole image encoder can definitely be a viable option. However, it requires a huge training time, especially when the training dataset is so large, this is a big issue in general for research being conducted at Universities with limited resources. This expensive training had also been an issue with existing worldwide geo-localization methods. To minimize the training cost, we only train the added linear layers.
> * We have tried finetuning the CLIP backbone also. But we observed it did not perform better compared to the case where a frozen backbone is used. This may be the consequence of catastrophic forgetting of original CLIP weights while training, i.e., the knowledge of a huge training dataset of the CLIP foundation model (400 million image-text pairs) stored in the form of learned weights gets lost while updating them. Based on experimental observations, keeping the frozen CLIP backbone and only training linear layers (h1 and h2) was found to better adapt to our image-to-GPS retrieval task.
> * Another benefit of using pretrained CLIP weights is that they can be precomputed, i.e., the CLIP features on the training dataset can be saved beforehand. We observed doing this drastically reduced our training time. For instance, the training time of one epoch on MP-16 dataset was reduced substantially from $\approx27$hrs to $\approx15$ minutes.
> * As CLIP is a multimodal architecture (its image and text encoders are aligned), when we align the pretrained image encoder of CLIP with our location encoder, the pretrained text encoder of CLIP also gets implicitly aligned with the location encoder (Lines 325-327). As a result, without separate training on text, our method, GeoCLIP can also process text queries enabling geo-localization using text.
>
> **Sampling of gallery GPS coordinates**: The traditional image-to-image retrieval methods use a reference gallery comprising images from the test set. In our case, if we follow the same protocol, i.e., creating a reference gallery with the GPS coordinates of the test set, then it will become an easier task. For instance, in the case of the Im2GPS3k test dataset, the number of GPS coordinates would be 3000. Our method GeoCLIP achieves strong localization performance via image-to-GPS retrieval in such cases. Below we present the results when the gallery of GPS coordinates is formed using a test set.
>
> |  Dataset |   1km  |  25km  |  200km |  750km | 2500km |
> |:--------:|:------:|:------:|:------:|:------:|:------:|
> | Im2GPS3k | 29.66 | 47.64 | 59.05 | 72.53 | 85.78 |
>
> The experimental results reported in the paper were performed on a more difficult task. The training set MP-16 dataset contains 4.72 million images. Sampling 100k or 500k samples from it to construct reference gallery GPS coordinates requires a test image to be searched on a larger gallery size which is tougher than searching on a small gallery obtained using a test set.
>
> Even if the test set contains images that are out of distribution (different from the training data distribution), our method still performs reasonably well. For instance, the recent Google World Streets 15K (GWS15k) is a challenging dataset [5], with locations that have either not been densely covered in the training dataset or that are at a considerable distance from covered regions. As a consequence, existing methods do not perform well on this dataset. Our method comfortably beats state-of-the-art methods on it (Table 1(b)).
>
> The reason for creating a reference gallery of GPS coordinates using the training set is to utilize prior knowledge of the dataset. This knowledge helps to restrict our search to regions where pictures are mostly taken, and to narrow down our search to geographic places of interest. Even classification-based methods divide the globe into discrete, non-uniform geographic cells (classes) by using the knowledge of the training set. However, our method can also achieve decent performance without using such a dataset prior. This is shown in the results below, where we uniformly sample across the Earth or sample evenly spaced coordinates using a Fibonacci sphere.
>
> |            Sampling           | 1km  |  25km | 200km | 750km | 2500km |
> |:-----------------------------:|:----:|:-----:|:-----:|:-----:|:------:|
> | Uniform  (1 M, Land Only)   | 0.70 | 11.95 | 37.04 | 59.09 |  78.14 |
> | Evenly Spaced (1M Land Only ) | 0.02 | 12.85 | 37.07 | 59.39 |  77.78 |
> | Uniform  (1 M All Earth)   |    0.10  |  7.17     |   32.03    |    54.28   |   75.17     |
> | Evenly Spaced  (1M All Earth) | 0.03 |  9.18 | 33.47 | 55.32 |  75.34 |
>
> **Negative societal impact**: Thank you for the suggestion. In our current draft, we had briefly mentioned avoiding unethical use that could lead to misleading detections or violate privacy (Lines 372-373). We will include the reviewer's suggested points to further strengthen our broader impacts section.

---

> > ### Comment · Reviewer_AAT4 · 2023-08-18
> >
> > I'd like to thank the authors for their detailed in well argumented rebuttal. I have read the other reviews and author rebuttals.
> > I think the authors have addressed the weaknesses raised by the reviewers, especially about the construction of the gallery and it size, well and provided insightful additional experiments.
> > All reviewers agree towards an acceptance of the paper and I will keep my initial rating. Similar to reviewer SMLq I encourage the authors to include the additional results and discussion wrt. to gallery size/construction in the final version of the paper and/or sup. material.

---

> > > ### Author Response · Authors · 2023-08-21
> > >
> > > Thank you for your thoughtful review and for recognizing the efforts we've put into addressing the weaknesses identified by the reviewers. We genuinely appreciate your support and your decision to maintain the initial rating. We are committed to improving the manuscript based on the feedback provided by all reviewers. As suggested, we will certainly include the additional results and discussions concerning the gallery size and construction in the final version of the paper, ensuring that the comments and observations provided are appropriately integrated.

---

### Official Review · Reviewer_UZhc · 2023-07-06

**Soundness:** 3 good
**Presentation:** 4 excellent
**Contribution:** 3 good
**Rating:** 7
**Confidence:** 5

**Summary:**

This paper proposes an image-to-GPS retrieval method, with an image encoder from the pre-trained CLIP and a tailored GPS encoder.

The GPS encoder is implemented by using:  i) the Equal Earth Projection to mitigate the GPS distortion; ii) Random Fourier transform with multiple MLPs to obtain high-frequency `positional encoding'.

Experiments on standard benchmarks demonstrate the effectiveness of the proposed method.

**Strengths:**

1) This paper is well-written and easy to follow;

2) The idea of this paper is interesting and new;

3) The experimental evaluations are strong, particularly the interesting Section 4.4.

**Weaknesses:**

I only have some minor comments:

1) Though it is interesting to adopt the CLIP pre-trained encoders, I would like to the experiments substituting the CLIP image encoder with Swin encoder (as GeoDecoder [5]), or substituting the Swin encoder in [5] with the CLIP image encoder.

This would help readers to clearly understand the contribution of the CLIP image encoder.

2) I would like to see an experiment under the following setting:

a) Evenly partition the earth into patches, and use the patch center GPS as the database;

b) perform the proposed image-to-GPS retrieval and report the performance;

c) Repeat a) and b) with coarse-to-fine patch partitions.

3) Please explicitly list the image-encoding time and the image-to-GPS retrieval time;

4) Please tone down the claim of "However, this approach is not practical as it becomes infeasible to construct a gallery containing images covering the entire world." The reason is that we have a world satellite image gallery.

**Questions:**

Please refer to weaknesses.

**Limitations:**

Yes.

---

> ### Author Rebuttal · Authors · 2023-08-08
>
> We thank the reviewer for the encouraging feedback on our section 4.4 and for finding our work to be “well-written and easy to follow”, and our idea to be “interesting and new”. Also, we are thankful to the reviewer for mentioning - “experimental evaluations are strong”. We reply to the queries below.
>
> **Contribution of CLIP image encoder**: We adopted CLIP pretrained encoder as it has been trained on 400 million (image, text) pairs and has shown strong generalization performance across different downstream tasks. Besides this, the muli-modal model architecture allowed us also to process text queries. The CLIP text features get inherent alignment with our location encoder, enabling us also to perform worldwide geo-localization using text queries. As suggested, we performed an experiment by substituting our CLIP image encoder with the Swin encoder, whose results are presented below (please note that the end-to-end training on large-scale geo-localization training dataset (MP-16) is computationally expensive and time consuming, and the experiment is still running).
>
> |     Im2GPS3k    | 1km |  25km | 200km | 750km | 2500km |
> |:---------------:|:---:|:-----:|:-----:|:-----:|:------:|
> | Swin (3 epochs) | 4.9 | 17.65 | 27.76 | 43.54 |  64.4  |
>
> **Results on coarse-to-fine patch partitions**: As suggested, we evenly partitioned the earth into patches using the Fibonacci sphere and subsequently used patch centers as GPS coordinates. Our coarse partitioning contained 10k patches, medium partitioning had 100k patches and the fine-grained consisted of 1M patches. Apart from evenly partitioning the whole earth, we also performed the experiments by evenly partitioning the land areas. Below we present the results of our image-to-GPS retrieval approach (GeoCLIP) on two different datasets (Im2GPS3k and GWS15k).
>
> |            Im2GPS3k           |  1km |  25km | 200km | 750km | 2500km |
> |:-----------------------------:|:----:|:-----:|:-----:|:-----:|:------:|
> | Coarse Patch  (10K All Earth) | 0.00 |  2.57 |  19.0 | 42.08 |  64.53 |
> | Medium Patch (100K All Earth) | 0.00 |  3.27 | 25.53 | 50.18 |  71.91 |
> |   Fine Patch (1M All Earth)   | **0.03** | **9.18** | **33.47** | **55.32** |  **75.34** |
> | Coarse Patch  (10K Land Only) | 0.00 |  2.14 | 20.12 | 45.65 |  68.17 |
> | Medium Patch (100K Land Only) | 0.00 |  7.24 | 30.16 | 53.99 |  74.64 |
> |   Fine Patch (1M Land Only)   | **0.02** | **12.85** | **37.07** | **59.39** |  **77.78** |
>
> |             GWS15k            |  1km | 25km | 200km | 750km | 2500km |
> |:-----------------------------:|:----:|:----:|:-----:|:-----:|:------:|
> | Coarse Patch  (10K All Earth) | 0.00 | 0.35 |  9.30 | 37.16 |  72.08 |
> | Medium Patch (100K All Earth) | 0.00 | 0.48 | 11.54 | 40.52 |  73.63 |
> |   Fine Patch (1M All Earth)   | **0.00** | **1.06** | **12.86** | **41.75** | **73.59** |
> | Coarse Patch  (10K Land Only) | 0.00 | 0.66 | 10.99 | 40.35 |  73.59 |
> | Medium Patch (100K Land Only) | 0.00 | 0.73 | 13.16 | 43.63 |  74.99 |
> |   Fine Patch (1M Land Only)   | **0.00** | **1.12** | **13.89** | **43.36** |  **74.91** |
>
> We observe better performance on the fine patch partitions across datasets, as it allows finer search on the GPS embeddings leading to better localization performance.
>
> **Image-encoding time and image-to-GPS retrieval time**: The time taken (in seconds) for encoding image and overall time for image-to-GPS retrieval (sum of image encoding time, GPS encoding time, and search time) is shown below for different gallery sizes on Im2GPS3k dataset.
>
> |  GPS Gallery |   Batch of 1 image  |                             | Batch of 128 images |                             |
> |--------------|:-------------------:|:---------------------------:|:-------------------:|:---------------------------:|
> |              | Image encoding Time | Image-to-GPS retrieval time | Image encoding Time | Image-to-GPS retrieval time |
> |     10k      |       0.00033       |            0.0006           |       0.06000       |            0.0674           |
> |     100k     |       0.00033       |            0.0033           |       0.06042       |            0.0745           |
> |     500k     |       0.00033       |            0.0152           |       0.06049       |            0.1043           |
>
> When the gallery size is varied, the image encoding time remains almost the same, while image-to-GPS retrieval time shows an increasing trend with the increase in gallery size.
>
> **Tone down the sentence**: Thank you for pointing this out. We will edit the sentence accordingly in the final version.

---

> > ### Comment · Reviewer_UZhc · 2023-08-17
> >
> > Thanks for the rebuttal and new experiments.
> >
> > In summary, this paper is interesting and I would maintain my initial rating.
> >
> > I would expect authors to include these new experiments in the paper/supplementary, and discuss the benefit of using CLIP.
> >
> > From the rebuttal, I can see that replacing the CLIP encoder with Swin encoder (as GeoDecoder [5]) would result in inferior performance. When referring to Table (a) in section 4.1, the numbers of [5] are 12.8/33.5/45.9/61.0/76.1.

---

> > > ### Author Response · Authors · 2023-08-18
> > >
> > > We sincerely thank the reviewer for taking the time to thoroughly assess our rebuttal responses, appreciating our work, and finding it interesting. It is very encouraging for us to know that the reviewer is maintaining the rating. As suggested by the reviewer, we will make sure to incorporate the new experimental results and a comprehensive discussion on the merits of CLIP into the final manuscript.

---

### Official Review · Reviewer_AH57 · 2023-07-07

**Soundness:** 3 good
**Presentation:** 3 good
**Contribution:** 3 good
**Rating:** 5
**Confidence:** 4

**Summary:**

This work learns an image-to-GPS retrieval approach where the image is a query, and the gallery
is a collection of GPS coordinates. The authors propose GeoCLIP, which consists of a location encoder, and an image encoder. The retrieval then is based given a new query image against a collection of gps embeddings. Experimental results show that this work outperforms previous SOTA at the Im2GPS3k and GWS15k datasets, across all distance thresholds.

**Strengths:**

* This is a well written paper, with good motivation, writing and experimental section.
* The extra results on "Utility of our location encoder beyond geo-localization" provide some more arguments about the effectiveness of their location encoder
*  The discussion about encoding GPS with an MLP is very interesting and is backed by good experiments.
 * Hierarchical learning experiments are solid and seem to provide good justification for the design choices.

**Weaknesses:**

* For Figure 3, I have no baseline to compare against. How would other methods compare in terms of how fast the curve degrades?
* For the GPS encoding, how come the "city" scale is the one with the biggest rise? Could it be something related to the choice of parameters for the Random Fourier Features? It might make sense if some of the discussion from the supplementary material can be transferred to the main paper here since this is a key part of the method.
* For the hierarchical method, how much does the merging of the outputs from the different encoders affect the overall computational needs?

**Questions:**

I have stated my questions above in the weaknesses, so this is what I would like to get answers from the authors on.

**Limitations:**

The authors do address this in their work.

---

> ### Author Rebuttal · Authors · 2023-08-08
>
> We thank the reviewer for the positive feedback on our work: “well-written paper, with good motivation”, “hierarchical learning experiments are solid”, and “good justification for design choices”. We are also thankful to the reviewer for the encouraging remarks about our section ‘Utility of our location encoder beyond geo-localization’ and for appreciating our discussion and experiments on GPS embedding with MLP. We reply to the queries below.
>
> **Figure 3 (baseline to compare)**: In Figure 3, we showed the efficacy of our proposed method GeoCLIP in limited data settings where we observed respectable performance even with 5% training data. As suggested, we performed the same experiment with the classification baseline approach (ISNs [12]). We summarize the comparison of our method with the classification approach in the table below. The value inside the brackets on a particular row shows the performance degradation compared to the respective method’s performance with 100% training data.
>
> | Im2GPS3k                | **100 %** |    **20%**   |    **10%**    |    **5%**    |
> |-------------------------|:---------:|:------------:|:-------------:|:------------:|
> | 2500 km (GeoCLIP, Ours) |    83.8   |  83.8 (-0.0) |  83.6 (-0.2)  |  83.2 (-0.6) |
> | 2500km  (ISN [12])      |    66.0   | 50.1 (**-15.9**) |  48.7 (**-17.3**) | 43.9 (**-22.1**) |
> | 750km (GeoCLIP, Ours)   |    69.7   |  69.2 (-0.5) |  69.1 (-0.6)  |  67.6 (-2.1) |
> | 750km  (ISN [12])       |    49.7   | 31.9 (**-17.8**) | 29.2 (**-20.5**)) | 25.1 (**-24.6**) |
> | 200km (GeoCLIP, Ours)   |    50.6   |  49.0 (-1.6) |  48.9 (-1.7)  |  48.7 (-1.9) |
> | 200km  (ISN [12])       |    36.6   | 20.0 (**-16.6**) |  17.4 (**-19.2**) | 14.3 (**-22.3**) |
> | 25km (GeoCLIP, Ours)    |    34.5   |  32.6 (-1.9) |  31.9 (-2.6)  |  31.1 (-3.4) |
> | 25km  (ISN [12])        |    28.0   | 14.1 (**-13.9**) |  11.9 (**-16.1**) |  9.6 (**-18.4**) |
> | 1km (GeoCLIP, Ours)     |    14.1   |  13.1 (-1.0) |  12.1 (-2.0)  |  10.8 (-3.3) |
> | 1km  (ISN [12])         |    10.5   |  5.1 (**-5.4**)  |   4.5 (**-6.0**)  |  3.2 (**-7.3**)  |
>
> It can be easily observed that the classification method's performance degrades much faster than our approach when the amount of training data is reduced, highlighting our method to be more data-efficient.
>
> **Performance improvement in city scale**: The embeddings of 2D GPS coordinates obtained using direct MLPs suffer from spectral bias, i.e., they only capture low-frequency information and fail to capture high-frequency details (Lines 54-59, 162-164). Thus, their performance (Table 2a first row) is good only on large-scale distance thresholds (2500 km). Particularly on the small scale distance thresholds such as street and city, require features to capture fine-grained high-frequency details. MLP performance is very poor on them. To improve the performance on smaller scales, we applied Random Fourier Features (RFF).
>
> We found $\sigma$ to be an important parameter in RFF (eq. 2 and 3), which can control the frequency range. As shown in Figure 2 of the supplementary, the small $\sigma$ value performs well on coarser granularity (200km, 750km, and 200 km), while the large $\sigma$ performs well on fine-grained levels (1km and 25 km). One intuitive way to overcome this trade-off is to select an intermediate $\sigma$ value ($2^4$) which turned out to perform very well on a city scale (Table 2(a)) compared to the MLP baseline. Yes, the performance improvement can be attributed to the choice of a particular $\sigma$ value (parameter of RFF). Moreover, our hierarchical strategy provides further gains on all distance threshold metrics (last row of Table 2(b)) compared to the performance obtained with the individual best $\sigma$ value (last row of Table 2(a)). As suggested, we will add more discussion in the main paper to avoid any confusion.
>
> **Merging outputs of different encoders of hierarchical method**: We merge the GPS embeddings by simple addition operation (Lines 194-195). These are constant operations performed on M different vectors for M level hierarchy, which is fixed across the experiments on different datasets, maintaining a similar time complexity as any individual encoder. Besides this, our location encoder is just a small-sized shallow network containing MLPs of 4 hidden layers and one output layer, which is computationally lightweight compared to popular deep network architectures.

---

### Official Review · Reviewer_vH7S · 2023-07-08

**Soundness:** 3 good
**Presentation:** 3 good
**Contribution:** 3 good
**Rating:** 6
**Confidence:** 4

**Summary:**

For global localization task for a query photo, the authors propose a CLIP-inspired image-to-GPS retrieval approach (named ‘GeoCLIP’) where we retrieve the GPS coordinates of an unseen query image by matching it with the gallery of GPS coordinates.  Regarding location encoder, some new ideas such as Random Fourier Features and Hierarchical Representation were introduced. The experimental results showed the effectiveness of the proposed method by outpeforming the existing baselines.


**Strengths:**

* Matching GPS location embeddings and image embeddings (GeoCLIP) are introduced into an image localization problem. This is novel.

* It is very interesting that the location encoder gets inherent alignment with CLIP’s text features, enabling us to map textual descriptions to geographical coordinates.

* The proposed location encoding method employing Random Fourier Features is much improved compared with the straightforward method which is the direct use of an individual MLP to encode GPS coordinates into feature vectors.

* The experimental results showed that the proposed method outperformed the existing baselines which employs image search methods or grid classification methods.

**Weaknesses:**

* Matching an image embedding with many location embedding (100K (Im2GPS3k) and 500K (GWS15k) coordinates) is needed to estimate a GPS location.


**Questions:**

* This approach is interesting. However, it seems to assume that one location corresponds to one image feature. Is it always true ?  For example, in restraurants, there are many kinds of meal menus. In some places, the view might be changed greatly with viewing direction change or even with a small location movement.

* When searching the location of a query image, matching it with a collection of GPS coordinates is needed. The needed gallery size of the GPS coordinate seems to be large. How long does it take to estimate the location of one image in case of 100K (Im2GPS3k) and 500K (GWS15k) coordinates.? It seems to need a relatively long time. How about comparison to the baselines regarding search time ?
Of course, all search-based methods has this disadvantage as well. However, for image search, many fast search methods has been proposed such as hash-based search. Is it possible to use any fast search methods for the proposed method ?
How about search time and computational cost compared with the classification-based methods ?


**Limitations:**

A limitation was mentioned regarding image encoding at times of training.

---

> ### Author Rebuttal · Authors · 2023-08-08
>
> We thank the reviewer for finding our method GeoCLIP (“matching of GPS location embeddings and image embeddings”) as “novel”, for appreciating our location encoder having “new ideas such as Random Fourier Features and Hierarchical Representation”, and also for mentioning our method’s ability to “map textual descriptions to geographical coordinates” as “interesting”. We reply to the queries below.
>
> **One location corresponding to one image feature**: We agree that the view can change drastically by changing the viewing direction. In traditional Geo-localization literature formulation, an image is assigned the GPS location of the camera with which the picture was taken. If the GPS of the camera taking the picture does not change regardless of viewpoint, the same GPS is assigned. We follow the traditional formulation, which is also followed by the existing works. Thus, same GPS location can have images of different views which might have different CLIP embeddings. That’s why instead of directly using CLIP embeddings, we add two trainable linear layers on top of our CLIP image backbone. Thereby, enforcing different views to match the same GPS embedding via our trainable layers, the image encoder will inherently learn view invariant features corresponding to one location.
>
> **Time taken to estimate the location of one image in case of 100K (Im2GPS3k) and 500K (GWS15k) coordinates**: Below, we provide the location estimation time (in seconds).
> | GPS Gallery           | Time per Batch (1 image) | Time per Batch (128 images)|
> |-----------------------|------------------|---------------------|
> | 100k                  | 0.0033            | 0.0745               |
> | 500k                  | 0.0152       | 0.1043               |
>
> **Comparison of search time with baselines**: The traditional image-to-image retrieval matches query image embedding with the reference gallery of image embeddings. We also use an image as a query, but our reference gallery consists of GPS embeddings. The time taken to obtain the image embeddings for the query remains the same in both approaches (image-to-image and image-to-GPS). Also, our location encoder employs the same size of GPS embedding vector as the size of image embedding. The gallery embeddings in the image-to-image retrieval approach require all the gallery images to pass through the image encoder, which can be a compute-intensive process. However, the time taken to get embeddings for the reference gallery is significantly lower in our method, as our location encoder is a lightweight network containing MLPs of 4 hidden layers and one output layer. Once the embeddings are obtained, search time would remain the same in both cases.
>
> **Fast search methods**: We can use any search algorithm to match the query image embedding with the reference gallery GPS embeddings. Yes, any fast search method can also be used.
>
> **Comparison with classification-based methods**: The classification methods divide the globe into discrete geographical cells (used as classes), where an image is classified to identify which cell it belongs to. Thereby the center coordinates of the geographic cell are used as the prediction for the GPS location of the query image. Though they can obtain fast predictions on a given query, the classification strategy results in coarse predictions (the predefined classes limit localization performance), and classes have to be pre-defined during training. On the other hand, the retrieval-based methods match an input query with the geo-tagged gallery images, and the GPS of the matching gallery image is assigned to the query image. We directly match the input query image with the gallery of GPS coordinates. Even though the prediction time would be higher compared to classification, retrieval-based methods can be applied to any size gallery, and the gallery can be updated anytime without re-training, and also, these methods enable fine-grain geo-localization.

---

> > ### Comment · Reviewer_vH7S · 2023-08-21
> >
> > Thanks for the response. I have read the other reviews and authors' rebuttals. As results, most of the reviewers' concerns seems to have been resolved. All reviewers are in agreement. I would like to keep my initial rating.

---

### Author Rebuttal · Authors · 2023-08-08

We sincerely thank all the reviewers for their valuable and encouraging feedback on our work. We are happy to see that all the reviewers have appreciated our work. The reviewers vH7S, AAT4, and SMLq even explicitly mentioned our approach to be novel. Moreover, we are very thankful that our demonstrated applications were well appreciated by reviewers vH7S, AH57, UZhc, and SMLq. It is also very encouraging that SMLq mentions that our work has the potential to benefit the wider community. We reply to the queries of each reviewer individually. We also attach a pdf containing figures, which we appropriately refer to while answering specific questions.

---

### Decision · Program_Chairs · 2023-09-21

**Decision:**

Accept (poster)

**Comment:**

All reviewers unanimously recommend acceptance, claiming that the papertackles an important problem, has technical novelty and presents extensive and compelling evaluations. The paper further passed an ethics review, that also deemed the paper fit for publication after the authors add some discussion on defense strategies against actors misusing the method.

The authors should take into account all the feedback from technical and ethics reviewers when preparing their final version.